# CYCLIC REPRESENTATIONS OF $U_q(\widehat{\mathfrak{sl}}_2)$ AND ITS BOREL SUBALGEBRAS AT ROOTS OF UNITY AND Q-OPERATORS

ROBERT WESTON

ABSTRACT. We consider the cyclic representations $\Omega_{rs}$ of $U_q(\widehat{\mathfrak{sl}}_2)$ at $q^N = 1$ that depend upon two points $r, s$ in the chiral Potts algebraic curve. We show how $\Omega_{rs}$ is related to the tensor product $\rho_r \otimes \bar{\rho}_s$ of two representations of the upper Borel subalgebra of $U_q(\widehat{\mathfrak{sl}}_2)$. This result is analogous to the factorization property of the Verma module of $U_q(\widehat{\mathfrak{sl}}_2)$ at generic-$q$ in terms of two q-oscillator representation of the Borel subalgebra - a key step in the construction of the Q-operator. We construct short exact sequences of the different representations and use the results to construct Q operators that satisfy TQ relations for $q^N = 1$ for both the 6-vertex and $\tau_2$ models.

## 1. INTRODUCTION

1.1. **Background.** In 1990, Bazhanov and Stroganov published a remarkable paper [BS90] in which they established a connection between two very different models: the six-vertex model and the chiral Potts model. This connection was put into a more algebraic framework and generalized in the papers [DJMM91a, DJMM91b, DJMM91c, BKMS90]. The key observation is that when $q^N = 1$ the algebra $U_q(\widehat{\mathfrak{sl}}_2)$ possesses $N$-dimensional so called *cyclic* representations in addition to the standard finite-dimensional representations inherited from $sl_2$. The cyclic representations $\Omega_{rs}$ are defined in Section 2 of the current paper and depend upon a pair of points $(r, s) \in \mathcal{C}_k \times \mathcal{C}_k$, where $\mathcal{C}_k$ is a complex algebraic curve. The R-matrix $\check{R}(rr'; ss')$ corresponding to the $U_q(\widehat{\mathfrak{sl}}_2)$ isomorphism $\Omega_{rr'} \otimes \Omega_{ss'} \to \Omega_{ss'} \otimes \Omega_{rr'}$ was found to factorize into the composition of four $U_q(\widehat{\mathfrak{sl}}_2)$ isomorphisms, each of which swaps just one pair of the four indices $r, r', s, s'$ (see Proposition 2.1). These four factors encode the Boltzmann weights of the chiral Potts model.

The above work, and in particular the R-matrix factorization property, has been very influential in the field of quantum integrable systems. It became the inspiration for many papers on the factorization of R-matrices, L-operators and transfer matrices, for example [BLZ96, BLZ97, Der08, DKK06, Der07, BJM+09, BLMS10]. The most significant of these subsequent results was the algebraic construction of Baxter's Q-operator and an understanding of how transfer matrices factor into products or combinations of products of Q-operators [BLZ96, BLZ97]. A nice description of the algebraic construction of Q-operators and extensive references to the original literature can be found in the book [JMS21].

The key to understanding the algebraic construction and properties of the Q-operator for $U_q(\widehat{\mathfrak{sl}}_2)$ (generated by $e_i, f_i, t_i^{\pm 1}, i \in \{0, 1\}$) with $q$ generic is to consider infinite-dimensional representations of the Borel subalgebra $U_q(\mathfrak{b}_+)$ generated by $e_i, t_i^{\pm 1}$ [BLZ96, BLZ97]. Adopting the notation of [CVW24] , there are precisely two independent infinite-dimensional $U_q(\mathfrak{b}_+)$ representations $\rho_z$ and $\bar{\rho}_z$ that are *not* restrictions of representations of the full algebra $U_q(\widehat{\mathfrak{sl}}_2)$. Two corresponding Q-operators, $\mathcal{Q}(z)$ and $\bar{\mathcal{Q}}(z)$ are defined as transfer matrices (i.e. twisted traces over the auxiliary

*Date*: January 7, 2025.

2020 *Mathematics Subject Classification.* Primary 81R10, 81R12, 81R50; Secondary 82B23, 16T25.

space of products of L-operators) by choosing the auxiliary spaces as $\rho_z$ and $\bar{\rho}_z$ respectively. In this story, there are also three other representations that may be chosen as auxiliary spaces: $\nu_z^{\mu}$ - an infinite-dimensional Verma module representation of $U_q(\widehat{\mathfrak{sl}}_2)$ that depends on a complex weight $\mu$; $V_z$ - the two dimensional evaluation representation of $U_q(\widehat{\mathfrak{sl}}_2)$; and $\phi_z$ - an infinite-dimensional representation of $U_q(\mathfrak{b}_+)$ which is *triangular* (meaning that $e_1$ acts non-trivially, but $e_0$ acts as 0). There are two relations between these different representations that give rise to corresponding properties of transfer matrices and Q-operators. They are:

(i) A $U_q(\mathfrak{b}_+)$ isomorphism    $\nu_z^{\mu} \otimes \phi_z \simeq \rho_{zq^{-\mu/2}} \otimes \bar{\rho}_{zq^{\mu/2}}$,

(ii) A short exact sequence    $0 \longrightarrow \rho_{zq} \xrightarrow{\iota} \rho_z \otimes V_z \xrightarrow{\tau} \rho_{zq^{-1}} \longrightarrow 0$
   in which $\iota$ and $\tau$ are $U_q(\mathfrak{b}_+)$ intertwiners.

Let us denote the transfer matrices associated with the auxiliary spaces $\nu_z^{\mu}$ and $V_z$ by $T_{\mu}(z)$ and $T(z)$ respectively. The triangularity of $\phi_z$ means that the corresponding L-operator is a triangular matrix, which in turn means that the transfer matrix with $\phi_z$ as the auxiliary space is diagonal (and in a given spin sector is proportional to the identity). Relations of the following form then follow directly from (i) and (ii) respectively (details and notation can be found in [VW20, CVW24]):

$$
\begin{align}
(1.1) \qquad\qquad T_{\mu}(z) &\propto \mathcal{Q}(zq^{-\mu/2})\bar{\mathcal{Q}}(zq^{\mu/2}), \\
(1.2) \qquad\qquad T(z)\mathcal{Q}(z) &= \alpha(z)\mathcal{Q}(zq) + \beta(z)\mathcal{Q}(zq^{-1}).
\end{align}
$$

where $\alpha(z)$ and $\beta(z)$ are rational functions of $z$. Relation (1.1) can be viewed as more fundamental because (1.2) can be derived from it by considering integer $\mu$ (at which points $\nu_z^{\mu}$ becomes reducible).

This algebraic picture is relatively simple in the $U_q(\widehat{\mathfrak{sl}}_2)$ case discussed here because the two $U_q(\mathfrak{b}_+)$ representations $\rho_z$, $\bar{\rho}_z$ can be constructed as q-oscillator representations. This is not the case for general higher-rank algebras. However, analogous infinite-dimensional *pre-fundamental* representations of Borel subalgebras were constructed in [HJ12] and used to extend the above picture to arbitrary untwisted quantum affine algebras. The triumphal success of this programme was to construct a full algebraic proof of the long-standing Frenkel-Reshetikhin conjectures about the form of Bethe Ansatz equations for all untwisted quantum affine algebras [FH15]. This algebraic approach has continued to develop, and is at the heart of much recent work on QQ systems and q-difference opers (see for example [FH24, FJM24]). The approach has also been partially extended to open systems in [FS15, BT18, VW20, Tsu21, CVW24].

1.2. **Main Results.** Given that much of this large body of work was at least inspired by the paper [BS90], it is surprising that a fuller explanation of the the $q^N$ cyclic case of [BS90] that uses the above algebraic language has not been developed (we mention work that *has* been done in later sections). In particular, the role of representations of the Borel subalgeba has not been explained. This paper seeks to fill this gap.

The main results of this paper are as follows: after summarising the definition and properties of the $U_q(\widehat{\mathfrak{sl}}_2)$ cyclic representations of [DJMM91b] in Section 2.1, we go on in Section 2.2 to introduce $U_q(\mathfrak{b}_+)$ cyclic representations $\rho_r$ an $\bar{\rho}_r$ for $q^N = 1$ as well as another $N$-dimensional $U_q(\mathfrak{b}_+)$ representation $\varphi_c$. The later is the analog of the triangular representation $\phi_z$ mentioned above, but is even simpler in that is a direct sum of one-dimensional representations (both $e_0$ and $e_1$ act as 0). We then have Theorem 3.3 that gives the $U_q(\mathfrak{b}_+)$ isomorphism

$$\Omega_{rs} \otimes \varphi_{c_0} \simeq \rho_r \otimes \bar{\rho}_s.$$

This is the analog of property (i) above. We then go on in Theorem 3.7 to give $U_q(\mathfrak{b}_+)$ short exact sequences that are analogous to (ii) above. Parallel results that give these factorization and fusion properties in terms of L-operators are then given in Propositions 3.13 and 3.14 respectively. We then go in Section 4 to use these various representations as auxiliary spaces in the definition of Q operators and transfer matrices.

The construction of Q depends on whether the quantum space (the space on which T and Q act) is the tensor product of 2-dimensional representations or of $N$-dimensional cyclic representations. In the former case, we use $\rho$ and $\bar\rho$ as the auxiliary spaces to construct operators $Q_\rho$ and $Q_{\bar\rho}$ respectively. These operators satisfy the factorization property of Proposition 4.2. This can be viewed as an analog of property (1.1), but now for the transfer matrix $T_\Omega$ associated with the auxiliary space cyclic representation $\Omega$. The two Q-operators also satisfy the standard 6-vertex model TQ relations, as expressed in Proposition 4.3 and and Equation (4.5).

In the case when the quantum space is the tensor product of $N$-dimensional cyclic representations, the Q-operator is defined in terms of an auxiliary space representation $\Omega$ by Equation (4.7), and the corresponding TQ relations are given by Proposition 4.4. This later result demonstrates and explains the fact that the Q-operator of the $\tau_2$ model is indeed related to the half-monodromy matrix of the chiral Potts model as first observed in [BS90].

1.3. **Goals and Motivation.** As we have indicated, the overall goal of the paper is to clarify and systemize the algebraic picture of the $q^N = 1$, cyclic, $U_q(\widehat{\mathfrak{sl}}_2)$ case, and in particular to understand the role of representations of the Borel subalgebra and how they may be used to construct Q-operators that satisfy TQ relations.

There are two key motivations behind the work. Firstly, by systemizing the approach as indicated we hope to open the way to developing the construction and properties of Q-operators for higher rank cases at $q^N = 1$. Secondly, we intend to use these results in order to construct Q-operators for open systems at $q^N = 1$, extending the generic-$q$ results of [CVW24] to a wider class of boundary conditions and to higher rank. As we explain in Section 5, there are reasons why it is simpler to develop such open system results in the $q^N = 1$ case first.

## 2. Cyclic Representations of $\widetilde{U}_q(\widehat{\mathfrak{sl}}_2)$ and $U_q(\mathfrak{b}_+)$

In this Section, we recall the cyclic representations of $\widetilde{U}_q(\widehat{\mathfrak{sl}}_2)$ defined in [DJMM91b] and their connection with the Boltzmann weights of the chiral Potts model. We shall then go on to define new cyclic representations of the Borel subalgebra $U_q(\mathfrak{b}_+)$, as well as a fully decomposable $U_q(\mathfrak{b}_+)$ representation $\varphi_c$. Throughout this paper we fix $q$ to be a primitive $N$'th root of unity, with $N$ odd and $N \geqslant 3$.

2.1. **Cyclic Representations and R-matrices of $\widetilde{U}_q(\widehat{\mathfrak{sl}}_2)$.** We closely follow the notation and conventions of Section 4 of [DJMM91b]. We begin by defining $\widetilde{U}_q(\widehat{\mathfrak{sl}}_2)$, which is an algebra over $\mathbb{C}$ generated by $e_i, f_i, t_i^{\pm 1}, z_i$ $(i = 0, 1)$, where $e_i, f_i, t_i^{\pm 1}$ satisfy the standard relations of the quantum affine algebra $U_q(\widehat{\mathfrak{sl}}_2)$, and $z_0$ and $z_1$ are two new central elements. The comultiplication of $\widetilde{U}_q(\widehat{\mathfrak{sl}}_2)$ is chosen as

$$\begin{aligned} \Delta(e_i) &= e_i \otimes \mathbb{I} + z_i t_i \otimes e_i, \quad \Delta(f_i) = f_i \otimes t_i^{-1} + z_i^{-1} \otimes f_i, \\ \Delta(t_i) &= t_i \otimes t_i, \quad \Delta(z_i) = z_i \otimes z_i. \end{aligned}$$

2.1.1. *Cyclic Representations $\Omega_{rs}$.* Let $W$ be an $N$-dimensional vector space over $\mathbb{C}$, and let $X, Z$ be invertible linear operators on $W$ satisfying the relations

$$(2.1) \qquad\qquad\qquad ZX = qXZ, \quad X^N = Z^N = \mathbb{I}.$$

Following [DJMM91b], we define cyclic representations $\Omega_{rs} : \widetilde{U}_q(\widehat{\mathfrak{sl}}_2) \to \mathrm{End}(W)$ that depend on a pair of points $(r, s) \in \mathcal{C}_k \times \mathcal{C}_k$. Here, $\mathcal{C}_k$ is the complex algebraic curve given by $(x, y, \mu) \in (\mathbb{C}^\times)^3$ such that

$$x^N + y^N = k(1 + x^N y^N), \quad \mu^N = \frac{k'}{1 - kx^N} = \frac{1 - ky^N}{k'},$$

where $k$ is the modulus with $k^2 + k'^2 = 1$. With the notation $r = (x_r, y_r, \mu_r)$ and $s = (x_s, y_s, \mu_s)$, the representation $\Omega_{rs} : \widetilde{U}_q(\widehat{\mathfrak{sl}}_2) \to \mathrm{End}(W)$ is given by [1]

(2.2)

$$\Omega_{rs}(e_0) = \frac{\kappa_0 x_r}{q - q^{-1}} X^{-1} \left( \frac{y_s}{x_r \mu_r \mu_s} Z^{-2} - 1 \right), \quad \Omega_{rs}(e_1) = \frac{\kappa_1 y_r}{q - q^{-1}} \left( \frac{x_s \mu_r \mu_s}{y_r} Z^2 - 1 \right) X,$$

$$\Omega_{rs}(f_0) = \frac{c_0 y_r}{q \kappa_0 x_r x_s (q - q^{-1})} \left( \frac{x_s \mu_r \mu_s}{y_r} Z^2 - 1 \right) X, \quad \Omega_{rs}(f_1) = \frac{c_0}{q \kappa_1 x_s (q - q^{-1})} X^{-1} \left( \frac{y_s}{x_r \mu_r \mu_s} Z^{-2} - 1 \right),$$

$$\Omega_{rs}(t_0) = \frac{c_0 y_r y_s}{q^2 x_r x_s \mu_r \mu_s} Z^{-2}, \quad \Omega_{rs}(t_1) = (\Omega_{rs}(t_0))^{-1}, \quad \Omega_{rs}(z_0) = c_0, \quad \Omega_{rs}(z_1) = \frac{1}{c_0},$$

where

(2.3)
$$c_0^2 = \frac{q^2 x_r x_s}{y_r y_s},$$

and the values of the fixed parameters $\kappa_0, \kappa_1$ can be found in [DJMM91b]. When it clarifies the expression we sometimes use the notation $\Omega_{r,s}$ instead of $\Omega_{rs}$.

2.1.2. *R-Matrices $\check{R}(rr'; ss')$.* The $\widetilde{U}_q(\widehat{\mathfrak{sl}}_2)$ representation $\Omega_{rs}$ was constructed, and parameterized in terms of $(r, s) \in \mathcal{C}_k \times \mathcal{C}_k$, precisely in order that an invertible $\widetilde{U}_q(\widehat{\mathfrak{sl}}_2)$ intertwiner

(2.4)                                    $$\check{R}(rr'; ss') : W \otimes W \to W \otimes W$$

existed with the property

(2.5)    $$\check{R}(rr'; ss')(\Omega_{rr'} \otimes \Omega_{ss'})\Delta(x) = (\Omega_{ss'} \otimes \Omega_{rr'})\Delta(x)\check{R}(rr'; ss'), \quad \text{for all } x \in \widetilde{U}_q(\widehat{\mathfrak{sl}}_2).$$

Of course, the pair of properties 2.4 and 2.5 is the same as the statement that there exists a $\widetilde{U}_q(\widehat{\mathfrak{sl}}_2)$ isomorphism $\check{R}(rr'; ss') : \Omega_{rr'} \otimes \Omega_{ss'} \to \Omega_{ss'} \otimes \Omega_{rr'}$.

The key factorization property of the R-matrix that was discussed in Section 1 is as follows.

**Proposition 2.1** (DJMMb). *The $\widetilde{U}_q(\widehat{\mathfrak{sl}}_2)$ isomorphism $\check{R}(rr'; ss') : \Omega_{rr'} \otimes \Omega_{ss'} \to \Omega_{ss'} \otimes \Omega_{rr'}$ can be written in terms of two $\widetilde{U}_q(\widehat{\mathfrak{sl}}_2)$ isomorphisms*

$$\begin{aligned} T_{rs} : \Omega_{rs} &\to \Omega_{sr}, and \\ S_{r's} : \Omega_{rr'} \otimes \Omega_{ss'} &\to \Omega_{rs} \otimes \Omega_{r's'}, \end{aligned}$$

*as follows*

(2.6)                                    $$\check{R}(rr'; ss') = S_{rs'}(T_{rs} \otimes T_{r's'})S_{r's},$$

*where*

(2.7)                $$T_{rs} = \sum_{n=0}^{N-1} \overline{W}_{rs}(n) Z^{2n}, \quad S_{rs} = \sum_{n=0}^{N-1} \widehat{W}_{rs}(n) \chi^n, \quad \chi := X^{-1} \otimes X,$$

---

[1]To avoid confusion, we note that our $\Omega_{rs}$ corresponds to $\xi = sr$ in the notation of [DJMM91b]. The other difference in notation is that $Z$ of [DJMM91b] is our $Z^2$ - the reason for this change will become apparent when we consider L-operators in Section 3.3.

*in which the coefficients $\overline{W}_{rs}(n)$ and $\widehat{W}_{rs}(n)$ are chosen to satisfy the recursion relations*

$$(2.8) \qquad \frac{\overline{W}_{rs}(n)}{\overline{W}_{rs}(n-1)} = \frac{\mu_r \mu_s (x_r q^2 - x_s q^{2n})}{y_s - y_r q^{2n}}, \qquad \frac{\widehat{W}_{rs}(n)}{\widehat{W}_{rs}(n-1)} = \frac{\mu_s y_r - \mu_r y_s q^{2(n-1)}}{\mu_s x_s - \mu_r x_r q^{2n}}.$$

*Proof.* The full proof is given in [DJMM91b]. Here, we note that assuming $T_{rs}$ and $S_{rs}$ are of the form (2.7) then it follows that they are $\widetilde{U}_q(\widehat{\mathfrak{sl}}_2)$ intertwiners of the form stated provided they satisfy respectively

$$(2.9) \qquad T_{rs}(y_r - x_s \mu_r \mu_s Z^2) X \;=\; (y_s - x_r \mu_r \mu_s Z^2) X T_{rs},$$

$$(2.10) \qquad \mu_r S_{rs}(Z^2 \otimes \mathbb{I})(x_r - y_s \chi) \;=\; \mu_s (Z^2 \otimes \mathbb{I})(x_s - y_r \chi) S_{rs}$$

These two conditions are then satisfied if the recursion relations (2.8) hold.

$\square$

It is useful to also consider discrete Fourier transforms of the coefficients $\widehat{W}_{rs}(n)$ and $\overline{W}_{rs}(n)$. Defining

$$(2.11) \qquad W_{rs}(n) = \sum_{m=0}^{N-1} \widehat{W}_{rs}(m) q^{-2mn}, \quad \widetilde{W}_{rs}(n) = \sum_{m=0}^{N-1} \overline{W}_{rs}(m) q^{2mn},$$

and using basic results about discrete Fourier transforms, the corresponding recursion relations become

$$(2.12) \qquad \frac{W_{rs}(n)}{W_{rs}(n-1)} \;=\; \frac{\mu_r}{\mu_s}\left(\frac{y_s - x_r q^{2n}}{y_r - x_s q^{2n}}\right),$$

$$(2.13) \qquad \frac{\widetilde{W}_{rs}(n)}{\widetilde{W}_{rs}(n-1)} \;=\; \frac{y_s - x_r q^{2n} \mu_r \mu_s}{y_r - x_s q^{2n} \mu_r \mu_s}.$$

The coefficients $\overline{W}_{rs}(n)$ and $W_{rs}(n)$ are the Boltzmann weights of the chiral Potts model [BPAY88], and Proposition 2.1 is the algebraic statement of the factorization property of the R-matrix first observed in [BS90]. The discrete Fourier transform $\widetilde{W}_{rs}(n)$ will be used in the proof in Appendix A.3. We fix the normalization of our weights by the choice

$$(2.14) \qquad \widetilde{W}_{rs}(0) = \widehat{W}_{rs}(0) = 1.$$

This normalization will determine the normalization of the coefficients $E_i$ that appear later in Proposition 3.15.

To help picture the various intertwiners, and for later usage it, we introduce a graphical notation for $S_{rs}$, $T_{rs}$ and $\check{R}_{rr';ss'}$. Representing $\Omega_{rs}$ by

$$\Omega_{rs} \sim \quad \genfrac{}{}{0pt}{}{s}{r} \!\!=\!\!=$$

we then use the following pictorial representations (with operators acting from West to East and North to South):

$$S_{r's} \sim \quad \text{[diagram]} \quad : \Omega_{rr'} \otimes \Omega_{ss'} \to \Omega_{rs} \otimes \Omega_{r's'}$$

$$T_{rs} \sim \quad \text{[diagram]} \quad : \Omega_{rs} \to \Omega_{rs}$$

We also define composite operators

$$
\begin{aligned}
\check{B}_{r';ss'} &= (\mathbb{I} \otimes T_{r's'})S_{r's} : \Omega_{rr'} \otimes \Omega_{ss'} \to \Omega_{rs} \otimes \Omega_{s'r'} \\
\check{A}_{rr';s} &= S_{rs}(T_{rr'} \otimes \mathbb{I}) : \Omega_{rr'} \otimes \Omega_{ss'} \to .\Omega_{r's} \otimes \Omega_{rs'}
\end{aligned}
$$

(2.15)

that we shall make use of in Section 4. Their graphical realization follows from that of $S$ and $T$ and is

$$\check{B}_{r';ss'} = (\mathbb{I} \otimes T_{r's'})S_{r's} = \quad \text{[diagram]} \quad : \Omega_{rr'} \otimes \Omega_{ss'} \to \Omega_{rs} \otimes \Omega_{s'r'}$$

$$\check{A}_{rr';s} = S_{rs}(T_{rr'} \otimes \mathbb{I}) = \quad \text{[diagram]} \quad : \Omega_{rr'} \otimes \Omega_{ss'} \to \Omega_{r's} \otimes \Omega_{rs'}$$

Hence we arrive at the following picture for the operator $\check{R}(rr'; ss')$:

$$\check{R}(rr'; ss') = \check{A}_{rs;s'r'} \circ \check{B}_{r';ss'} = \quad \text{[diagram]} \quad : \Omega_{rr'} \otimes \Omega_{ss'} \to \Omega_{ss'} \otimes \Omega_{rr'}$$

(2.16)

Readers familiar with the literature on the chiral Potts model may be wondering why we do not separate the close neighbour pairs of lines, use a picture for $R(rr'; ss)$ of the form

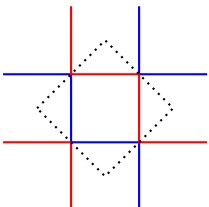

and then identify this picture directly with chiral Potts weights along the diagonal lines in the 'standard' way (see [BPAY88, BS90]). The answer is two fold: (i) separating the red and blue lines might suggest that $\Omega_{rs}$ directly factorizes into the tensor product of two representations, whereas the situation is more subtle as we shall see in the next section; (ii) the standard pictorial identification with diagonal chiral Potts weights $W_{rs}(n)$ and $\overline{W}_{rs}(n)$ corresponds to choosing $X$ as a diagonal matrix $X_{ij} = \delta_{i,j}q^{-2i}$ and $Z^2$ as a step operator $(Z^2)_{ij} = \delta_{i,\text{mod}(j+1,N)}$. This is allowed, but it is an unnatural choice from a representation theory point of view in that it would result in $\Omega_{rs}(t_i)$ acting as a step operator. More natural is the choice $X_{ij} = \delta_{i,\text{mod}(j+1,N)}$, $Z_{ij} = \delta_{i,j}q^i$, for $i, j \in \{0, 1, \cdots, N-1\}$. However, for now we make neither choice and leave $X$ and $Z$ as arbitrary $N \times N$ matrices satisfying (2.1).

2.2. **Cyclic Representations of $U_q(\mathfrak{b}_+)$.** Let us define the upper Borel subalgebra $U_q(\mathfrak{b}_+)$ to be the subalgebra of $\widetilde{U}_q(\widehat{\mathfrak{sl}}_2)$ generated by $e_i, t_i^{\pm 1}, z_i, i \in \{0, 1\}$. Then, following the spirit of the approach of [BLZ97] (which holds for for the generic-$q$ case with infinite-dimensional representations), we define two new cyclic Borel subalgebra representations $\rho_r, \bar{\rho}_r : U_q(\mathfrak{b}_+) \to \text{End}(W)$ that each depend on a single point $r \in \mathcal{C}_k$:

$$\rho_r(e_0) = -\frac{\kappa_0 x_r}{q - q^{-1}}X^{-1}, \quad \rho_r(e_1) = -\frac{\kappa_1 y_r}{q - q^{-1}}X, \quad \rho_r(t_0) = (q\mu_r)^{-1}Z^{-2}, \quad \rho_r(t_1) = q\mu_r Z^2,$$

$$\rho_r(z_0) = \rho_r(z_1) = 1,$$

$$\bar{\rho}_r(e_0) = \frac{\kappa_0 y_r}{\mu_r(q - q^{-1})}X^{-1}, \quad \bar{\rho}_r(e_1) = \frac{\kappa_1 x_r \mu_r}{q - q^{-1}}X, \quad \bar{\rho}_r(t_0) = \mu_r^{-1}Z^{-2}, \quad \bar{\rho}_r(t_1) = \mu_r Z^2,$$

$$\bar{\rho}_r(z_0) = \bar{\rho}_r(z_1) = 1$$

These two representations cannot be constructed by restricting full $\widetilde{U}_q(\widehat{\mathfrak{sl}}_2)$ representations. We need one final, $U_q(\mathfrak{b}_+)$ representation $\varphi_c : U_q(\mathfrak{b}_+) \to \text{End}(W)$, $c \in \mathbb{C}^\times$, defined by

$$\varphi_c(e_0) = \varphi_c(e_1) = 0, \quad \varphi_c(t_0) = \frac{c}{q}Z^{-2}, \quad \varphi_c(t_1) = \frac{q}{c}Z^2, \quad \varphi_c(z_0) = \frac{1}{c}, \quad \varphi_c(z_1) = c.$$

Clearly, $\varphi_c$ is the direct sum of $N$ one-dimensional $U_q(\mathfrak{b}_+)$ representations.

It is useful to extend our graphics to these new representations in order to have a more intuitive understanding of the results coming in Sections 3 and 4. To this end, let us representing $\rho_r, \bar{\rho}_r, \varphi_c$ by

$$\rho_r \sim \quad r \, \text{———} \qquad\qquad \bar{\rho}_r \sim \quad r \, \text{———} \qquad\qquad \varphi_c \sim \quad \text{- - - - -}$$

## 3. FACTORIZATION AND FUSION OF CYCLIC $U_q(\mathfrak{b}_+)$ REPRESENTATION

In this section we present a number of intertwiners associated with the representations introduced in Section 2. We will make use of these intertwiners in Section 4.

3.1. **Factorization.** We first define an operator $\mathcal{O}(\chi) \in \mathrm{End}(W \otimes W)$ as follows:

**Definition 3.1.** Let $\chi = X^{-1} \otimes X$, and let $\mathcal{O}(\chi) = \sum\limits_{n=0}^{N-1} a_n \chi^n$ be a polynomial with $a_n \in \mathbb{C}$ that satisfies the condition

$$(3.1) \qquad\qquad \mathcal{O}(q\chi) \;=\; \chi\, \mathcal{O}(q^{-1}\chi).$$

This condition determines $\mathcal{O}(\chi)$ up to a multiplicative constant, which we fix by the choice $a_n = q^{-n^2}$.

**Lemma 3.2.** *The operator $\mathcal{O}(\chi)$ has an inverse of the form $\mathcal{O}^{-1}(\chi) = \sum\limits_{n=0}^{N-1} b_n \chi^n$.*

The proof and form of the $b_n$ is given in Appendix A.1.

The representation $\Omega_{rs}$ defined in Section 2.1.1 is also of course a representation of the Borel subalgebra $U_q(\mathfrak{b}_+)$ and a key result of the current paper is the following Theorem 3.3. We call this result *factorization*.

**Theorem 3.3.** *We have the following $U_q(\mathfrak{b}_+)$ isomorphism*

$$\mathcal{O}(\chi) : \Omega_{rs} \otimes \varphi_{c_0} \to \rho_r \otimes \bar{\rho}_s,$$

*where $c_0$ is given in Equation (2.3).*

*Proof.* We first need to check the statement that

$$\mathcal{O}(\chi)(\Omega_{rs} \otimes \varphi_{c_0})(\Delta(x)) = (\rho_r \otimes \bar{\rho}_s)(\Delta(x))\mathcal{O}(\chi), \quad \text{for all } x \in U_q(\mathfrak{b}_+).$$

For $x = z_i$ the equality is clear, and for $x = t_i$ the relation follows from the property $[\chi, Z \otimes Z] = 0$ and the expression for $c_0$. For $x = e_0$, the relation becomes

$$\frac{\kappa_0 x_r}{q - q^{-1}} \mathcal{O}(\chi) \left( X^{-1} \left( \frac{y_s}{x_r \mu_r \mu_s} Z^{-2} - 1 \right) \otimes 1 \right) = \frac{\kappa_0}{q - q^{-1}}(-x_r X^{-1} \otimes 1 + \frac{y_s}{q\mu_r \mu_s} Z^{-2} \otimes X^{-1})\mathcal{O}(\chi)$$

which reduces to

$$q\, \mathcal{O}(\chi)(X^{-1} Z^{-2} \otimes 1) = (Z^{-2} \otimes X^{-1})\mathcal{O}(\chi).$$

This in turn becomes $\mathcal{O}(\chi)\, q\chi = \mathcal{O}(q^2 \chi)$, which follows from the defining property (3.1). Finally, for $x = e_1$ the relations becomes

$$\frac{\kappa_1 y_r}{q - q^{-1}} \mathcal{O}(\chi) \left( \left( \frac{x_s \mu_r \mu_s}{y_r} Z^2 - 1 \right) X \otimes 1 \right) = \frac{\kappa_1}{q - q^{-1}} \left( -y_r X \otimes 1 + q x_s \mu_r \mu_s Z^2 \otimes X \right) \mathcal{O}(\chi),$$

which reduces to

$$\mathcal{O}(\chi)(Z^2 X \otimes 1) = q(Z^2 \otimes X)\mathcal{O}(\chi),$$

which again follows from (3.1). The fact that $\mathcal{O}(\chi)$ is an isomorphism then follows from Lemma 3.2. $\qquad\square$

Again, it is useful to represent the operators $\mathcal{O}$ and $\mathcal{O}^{-1}$ and the associated isomorphisms graphically. We do this in the following way:

$$\mathcal{O} \sim \qquad\qquad\qquad : \Omega_{rs} \otimes \varphi_{c_0} \to \rho_r \otimes \bar{\rho}_s$$

$$\mathcal{O}^{-1} \sim \qquad\qquad\qquad : \rho_r \otimes \bar{\rho}_s \to \Omega_{rs} \otimes \varphi_{c_0}$$

*Remark* 3.4. We regard Proposition 3.3 as a root-of-unity, cyclic version of the analogous generic-$q$ theorem relating the Verma module to the tensor product of two infinite-dimensional Borel subalgebra representations. This latter result is at the heart of the Q-operator construction and can be found in varying forms throughout the literature including in [BLZ97, BŁMS10, BJM$^+$09, BGK$^+$14]. The statement of this result most resembling the form we have given can be found in [KT14]. Apart from the finite-dimensionality, one other difference between our root of unity case and the generic-$q$ case is that our $\varphi_c$ representation is a direct sum of 1-dimensional representations (both $e_0$ and $e_1$ act as zero), whereas in the generic-$q$ case its analog is a triangular representation (with one of $e_0$ or $e_1$ acting as zero and the other as a step operator).

Before going on to discuss short exact sequences associated with our different cyclic $U_q(\mathfrak{b}_+)$ representations, let us show how $\rho_r$ and $\bar{\rho}_s$ allow us to give a characterization of $T_{rs}$ and $S_{rs}$, and hence of chiral Potts weights, that is an alternative to the conventional picture of Proposition 2.1.

**Definition 3.5.** Let $\mathcal{P}(Z) = \sum\limits_{n=0}^{N-1} p_n Z^{2n}$ be a polynomial that satisfies the condition

$$(3.2) \qquad\qquad \mathcal{P}(Z) = Z^2 \mathcal{P}(q^{-1}Z).$$

The invertibility of $\mathcal{P}(Z)$ follows by a very similar argument to the proof given in Appendix A.1.

**Proposition 3.6.** *Defining* $\mathfrak{T}_{rs} = \mathcal{O}(\chi)(T_{rs} \otimes \mathbb{I})\mathcal{O}^{-1}(\chi)$, *and* $\mathfrak{S}_{rs} = (\mathcal{P}(Z)^{-1} \otimes \mathbb{I})S_{rs}(\mathcal{P}(Z) \otimes \mathbb{I})$, *we have the following* $U_q(\mathfrak{b}_+)$ *isomorphisms*

$$\begin{aligned}
(i) \quad \mathfrak{T}_{rs} \quad &: \quad \rho_r \otimes \bar{\rho}_s \to \rho_s \otimes \bar{\rho}_r, \\
(ii) \quad \mathfrak{S}_{rs} \quad &: \quad \bar{\rho}_r \otimes \rho_s \to \bar{\rho}_s \otimes \rho_r.
\end{aligned}$$

*Proof.* Statement $(i)$ follows as a simple corollary of Propositions 2.1 and 3.3. To prove (ii), we first note that we indeed have

$$\mathfrak{S}_{rs}\,(\bar{\rho}_r \otimes \rho_s)\Delta(t_i) = (\bar{\rho}_s \otimes \rho_r)\Delta(t_i)\,\mathfrak{S}_{rs}.$$

For commutation with $\Delta(e_0)$, we need

$$\frac{1}{\mu_r}\mathfrak{S}_{rs}\left(-y_r X^{-1} \otimes \mathbb{I} + x_s Z^{-2} \otimes X^{-1}\right) = \frac{1}{\mu_s}\left(-y_s X^{-1} \otimes \mathbb{I} + x_r Z^{-2} \otimes X^{-1}\right)\mathfrak{S}_{rs},$$

which can be re-expressed as

$$\frac{1}{\mu_r}S_{rs}\left(-y_r X^{-1}Z^{-2} \otimes \mathbb{I} + x_s Z^{-2} \otimes X^{-1}\right) = \frac{1}{\mu_s}\left(-y_s X^{-1}Z^{-2} \otimes \mathbb{I} + x_r Z^{-2} \otimes X^{-1}\right)S_{rs},$$

which is equivalent to the earlier condition (2.10). For $\Delta(e_1)$ the requirement for commutation is

$$\mu_r\mathfrak{S}_{rs}\left(-x_r X \otimes \mathbb{I} + y_s Z^2 \otimes X\right) = \mu_s\left(-x_s X \otimes \mathbb{I} + y_r Z^2 \otimes X\right)S_{rs},$$

which becomes

$$\mu_r S_{rs}\left(-x_r Z^2 X \otimes \mathbb{I} + y_s Z^2 \otimes X\right) = \mu_s\left(-x_s Z^2 X \otimes \mathbb{I} + y_r Z^2 \otimes X\right)S_{rs}$$

which can again be re-expressed as the condition (2.10). $\qquad\square$

3.2. **Fusion.** We now present three short exact sequences (SESs) for different $U_q(\mathfrak{b}_+)$ representations. In analogy with the classic results of [KRS81], we refer to these properties as *fusion*. The SESs also involve the standard two-dimensional evaluation representation $V_z : \widetilde{U}_q(\widehat{\mathfrak{sl}}_2) \to \operatorname{End} V$ ($z \in \mathbb{C}^\times$, $V = \mathbb{C}^2$) which we define by

(3.3)
$$
V_z(e_1) = \begin{pmatrix} 0 & z \\ 0 & 0 \end{pmatrix}, \qquad V_z(f_1) = \begin{pmatrix} 0 & 0 \\ z^{-1} & 0 \end{pmatrix}, \quad V_z(t_1) = \begin{pmatrix} q & 0 \\ 0 & q^{-1} \end{pmatrix}, \quad V_z(z_1) = 1,
$$
$$
V_z(e_0) = \begin{pmatrix} 0 & 0 \\ z & 0 \end{pmatrix}, \qquad V_z(f_0) = \begin{pmatrix} 0 & z^{-1} \\ 0 & 0 \end{pmatrix}, \quad V_z(t_0) = \begin{pmatrix} q^{-1} & 0 \\ 0 & q \end{pmatrix}, \quad V_z(z_0) = 1.
$$

Then we have the following:

**Theorem 3.7.** *Let* $r, s \in \mathcal{C}_k$, *with* $s = (x_s, y_s, \mu_s)$. *Defining* $sq^{\pm 1} = (x_s q^{\pm 1}, y_s q^{\pm 1}, \mu_s q^{\pm 1})$, *we see that we also have* $sq^{\pm 1} \in \mathcal{C}_k$. *Choose* $z_s$ *to satisfy* $z_s^2 = \kappa_0 \kappa_1 x_s y_s$ . *Let us define the following complex constants*

$$
c_s = -\frac{\kappa_0 x_s \mu_s q^2}{z_s}, \quad \bar{c}_s = \frac{\kappa_0 y_s q}{z_s}, \quad d_s = \frac{\kappa_0 y_s}{z_s q}.
$$

*Then we have the following* $U_q(\mathfrak{b}_+)$ *short exact sequences:*

$(i)$ $\qquad 0 \longrightarrow \rho_{sq} \xrightarrow{\iota_s} \rho_s \otimes V_{z_s} \xrightarrow{\tau_s} \rho_{sq^{-1}} \longrightarrow 0$ *where* $\iota_s = \begin{pmatrix} c_s Z \\ X Z^{-1} \end{pmatrix}$, $\tau_s = (-\dfrac{q}{c_s} X Z^{-1}, Z)$,

$(ii)$ $\qquad 0 \longrightarrow \bar{\rho}_{sq} \xrightarrow{\bar{\iota}_s} \bar{\rho}_s \otimes V_{z_s} \xrightarrow{\bar{\tau}_s} \bar{\rho}_{sq^{-1}} \longrightarrow 0$ *where* $\bar{\iota}_s = \begin{pmatrix} \bar{c}_s \\ X Z^{-2} \end{pmatrix}$, $\bar{\tau}_s = (-\dfrac{1}{\bar{c}_s} X Z^{-2}, 1)$,

$(iii)$ $\quad 0 \longrightarrow \Omega_{r,sq} \xrightarrow{\bar{I}_s} \Omega_{r,s} \otimes V_{z_s} \xrightarrow{\bar{T}_s} \Omega_{r,sq^{-1}} \longrightarrow 0$ *where* $\bar{I}_s = \begin{pmatrix} d_s \\ X \end{pmatrix}$, $\bar{T}_s = (-X, d_s)$.

*Proof.* Let us outline the proof of (i). The proofs of (ii) and (iii) are then very similar. First of all, the required $U_q(\mathfrak{b}_+)$ intertwining properties are equivalent to

$$
(\rho_s \otimes V_{z_s})(\Delta(x)) \circ \iota_s = \iota_s \circ \rho_{sq}(x), \quad \tau_s \circ (\rho_s \otimes V_{z_s})(\Delta(x)) = \rho_{sq^{-1}} \circ \tau_s, \quad \forall x \in U_q(\mathfrak{b}_+).
$$

These relations are just checked by explicit computation. The injectivity and surjectivity are of course equivalent to $\operatorname{Ker}(\iota_s)=0$, $\operatorname{Im}(\tau_s)=W$. These relations are also simple to check. Finally the SES property then follows from the fact that $\tau_s \circ \iota_s = -qX + ZXZ^{-1} = 0$.

$\square$

*Remark* 3.8. We note that while $\Omega_{rs}$ and $V_z$ appearing in $(iii)$ are also full $\widetilde{U}_q(\widehat{\mathfrak{sl}}_2)$ representations (unlike the $\rho_s$ and $\bar{\rho}_s$ representations of (i) and (ii)), $\bar{I}_s$ and $\bar{T}_s$ are only $U_q(\mathfrak{b}_+)$ intertwiners and are *not* $\widetilde{U}_q(\widehat{\mathfrak{sl}}_2)$ intertwiners.

Returning to the development of our graphical dictionary, we introduce the new pictorial representation

$$
V_z \sim \quad z \; \underline{\qquad}
$$

and then represent the injections and surjections involved in Proposition 3.7 as

$$\iota_s \sim \quad sq \, \underrightarrow{\quad}^{z_s}\, s \quad : \rho_{sq} \to \rho_s \otimes V_{z_s} \qquad\qquad \tau_s \sim \quad s \, \underrightarrow{\quad}^{z_s}\, sq^{-1} \quad : \rho_s \otimes V_{z_s} \to \rho_{sq^{-1}}$$

$$\bar{\iota}_s \sim \quad sq \, \underrightarrow{\quad}^{z_s}\, s \quad : \bar{\rho}_{sq} \to \bar{\rho}_s \otimes V_{z_s} \qquad\qquad \bar{\tau}_s \sim \quad s \, \underrightarrow{\quad}^{z_s}\, sq^{-1} \quad : \bar{\rho}_s \otimes V_{z_s} \to \bar{\rho}_{sq^{-1}}$$

$$\bar{I}_s \sim \quad {}^{sq}_{\ r} \, \underrightarrow{\quad}^{z_s}\, {}^{s}_{r} \quad : \Omega_{r,sq} \to \Omega_{r,s} \otimes V_{z_s} \qquad\qquad \bar{T}_s \sim \quad {}^{s}_{r} \, \underrightarrow{\quad}^{z_s}\, {}^{sq^{-1}}_{\ r} \quad : \Omega_{r,s} \otimes V_{z_s} \to \Omega_{r,sq^{-1}}$$

### 3.3. Factorization and Fusion of L-operators.

The factorization and fusion relations presented above also manifest themselves in terms of L-operators, and historically this is the form in which such relations have usually been discovered (see for example [BS90, BŁMS10]). The L-operators are also essential to consider as they are the building blocks of the T and Q operators defined in the Section 4.

If $\Pi : U_q(\mathfrak{b}_+) \to \mathrm{End}(W)$ denotes one of the four cyclic $U_q(\mathfrak{b}_+)$ representation defined in Section 2 (i.e., $\Pi$ is one of $\Omega_{rs}, \varphi_c, \rho_r, \bar{\rho}_r$), then the corresponding L-operator $\check{L}_\Pi(z) : W \otimes V \to V \otimes W$ is defined up to a normalization factor by the requirement

$$(3.4) \qquad \check{L}_\Pi(z)(\Pi \otimes V_z)\Delta(x) = (V_z \otimes \Pi)\Delta(x)\check{L}_\Pi(z) \quad \text{for all } x \in U_q(\mathfrak{b}_+).$$

In the case of $\check{L}_{\Omega_{rs}}(z)$, the relation (3.4) also extends to all $x \in \widetilde{U}_q(\widehat{\mathfrak{sl}}_2)$. As $\Pi \otimes V_z$ is generically irreducible as a $U_q(\mathfrak{b}_+)$ representation, this requirement fixes $\check{L}_\Pi(z)$ uniquely up to an arbitrary multiplicative constant. As is usual when discussing R-matrices and L-operators, it is also useful for us to consider the associated operator

$$(3.5) \qquad L_\Pi(z) = P\check{L}_\Pi(z), \quad \text{where} \quad P(a \otimes b) = b \otimes a.$$

Clearly we then have that $L_\Pi(z) \in \mathrm{End}(W \otimes V)$ with

$$(3.6) \qquad L_\Pi(z)(\Pi \otimes V_z)\Delta(x) = (\Pi \otimes V_z)\Delta^{op}(x)L_\Pi(z), \quad \text{for all } x \in U_q(\mathfrak{b}_+).$$

The 'rule' that we shall following for deciding which of $\check{L}_\Pi(z)$ or $L_\Pi(z)$ to use is that $\check{L}_\Pi$ is more natural when stating algebraic results and identifying with pictures, and $L_\Pi(z)$ easier for defining transfer matrices or Q-operators.

One other advantage of the $L_\Pi(z)$ version is that, since it is an element of $\mathrm{End}(W \otimes V)$, we can represent it as a $2 \times 2$ matrix with entries in $\mathrm{End}(W)$. In fact, all four $L_\Pi(z)$ have a common form. In order to express this form succinctly it is useful to introduce the following notation. Suppose $A$ and $B$ are $2 \times 2$ matrices with entries in $\mathbb{C}$. Then define $\{A, B\} \in \mathrm{End}(W \otimes V)$ as follows:

$$(3.7) \qquad \{A, B\} = \begin{pmatrix} X^{-1} & 0 \\ 0 & 1 \end{pmatrix} A \begin{pmatrix} Z^{-1} & 0 \\ 0 & Z \end{pmatrix} B \begin{pmatrix} X & 0 \\ 0 & 1 \end{pmatrix}.$$

Now let us define two matrices that depend on a point $r = (x, y, \mu) \in \mathcal{C}_k$ and $z \in \mathbb{C}^\times$ (with $\kappa_i$ the same constants as in Section 2):

$$(3.8) \qquad U_r(z) = \begin{pmatrix} z & \kappa_0 x\mu \\ \kappa_1 y & z\mu \end{pmatrix}, \quad V_r(z) = \begin{pmatrix} -qz & \kappa_0 y \\ q\kappa_1 x\mu & -z\mu \end{pmatrix}.$$

Using the notation (3.7), and after choosing a normalization, the four L-operators are given by

$$L_{\Omega_{rs}}(z) = \{U_r(z), V_s(z)\}, \quad L_{\varphi_c}(z) = \{\mathbb{I}, \mathbb{I}\},$$

(3.9)
$$L_{\rho_r}(z) = \{U_r(z), \mathbb{I}\}, \quad L_{\bar{\rho}_r}(z) = \{V_r(z), \mathbb{I}\},$$

where $\mathbb{I}$ is the 2×2 unit matrix. As $L_{\varphi_c}(z)$ is idependent of both $z$ and the constant $c$, we shall henceforth denote it as simply $L_\varphi$.

*Remark* 3.9. Explicit expressions for $L_{\Omega_{rs}}$ are not new - they appeared in [BS90] and have been used by various authors, for example in [Roa06, Nic10, MNP18]. The standard form (3.9) of L-operators that we use was inspired by the paper [MLP21] which introduces a similar form for an L-operator analogous to $L_{\Omega_{rs}}$ but for the case of *semi-cyclic* representations. The latter are discussed briefly in Section 5.

∅

In terms of the graphical notation we have already introduced, the four L-operators are given by the obvious pictures:

We will also need one other $\widetilde{U}_q(\widehat{\mathfrak{sl}}_2)$ isomorphism $\check{\mathbf{L}}_{\Omega_{rs}}(z) : V \otimes W \to W \otimes V$ that satisfies

(3.10)
$$\check{\mathbf{L}}_{\Omega_{rs}}(z)(V_z \otimes \Pi)\Delta(x) = (\Pi \otimes V_z)\Delta(x)\check{\mathbf{L}}_{\Omega_{rs}}(z) \quad \text{for all } x \in \widetilde{U}_q(\widehat{\mathfrak{sl}}_2).$$

We can clearly obtain this by inverting $\check{L}_{\Omega_{r,s}}(z)$, but we fix the normalization differently such that

$$\check{\mathbf{L}}_{\Omega_{rs}}(z)\check{L}_{\Omega_{r,s}}(z) = \check{L}_{\Omega_{r,s}}(z)\check{\mathbf{L}}_{\Omega_{rs}}(z) = q^2(z^2 - z_r^2)(z^2 - z_s^2)\mu_r\mu_s\,\mathbb{I},$$

where $z_r^2 = \kappa_0\kappa_1 x_r y_r$ as above. The explicit expression for this $\check{\mathbf{L}}_{\Omega_{rs}}(z)$ is given in Appendix A.3. We represent this operator graphically by

Our first result concerns the basic properties of $\check{L}_{\Omega_{r,s}}(z)$ which are the analogues of Proposition 2.1.

**Proposition 3.10.** *The L-operators defined by (3.9) satisfy the following relations:*

$$(i) \qquad (\mathbb{I} \otimes T_{rs})\,\check{L}_{\Omega_{rs}}(z) = \check{L}_{\Omega_{sr}}(z)\,(T_{rs} \otimes \mathbb{I}),$$

$$(ii) \qquad (\mathbb{I} \otimes S_{r's})\,\check{L}_{\Omega_{rr'}}(z) \otimes \check{L}_{\Omega_{ss'}}(z) = \check{L}_{\Omega_{rs}}(z) \otimes \check{L}_{\Omega_{r's'}}(z)\,(S_{r's} \otimes \mathbb{I}),$$

$$(iii) \qquad (\mathbb{I} \otimes \check{R}_{rr';ss'})\,\check{L}_{\Omega_{rr'}}(z) \otimes \check{L}_{\Omega_{ss'}}(z) = \check{L}_{\Omega_{ss'}}(z) \otimes \check{L}_{\Omega_{rr'}}(z)(\check{R}_{rr';ss'} \otimes \mathbb{I}).$$

*Proof.* We could prove each of these relations by using uniqueness of the associated $\widetilde{U}_q(\widehat{\mathfrak{sl}}_2)$ inter-twiner, but instead we take a more direct linear-algebra approach that exploits the simplicity of the simple form of the L-operators given by 3.9 (again, this approach is inspired by [MLP21]).

(i) Using the notation of (3.7) and (3.8), the statement (i) is equivalent to

$$T_{rs}(Z)\{U_r(z), V_s(z)\} = \{U_s(z), V_r(z)\} T_{rs}(Z),$$

which reduces to

$$\begin{pmatrix} T_{rs}(Zq^{-1}) & 0 \\ 0 & T_{rs}(Z) \end{pmatrix} U_r(z) \begin{pmatrix} Z^{-1} & 0 \\ 0 & Z \end{pmatrix} V_s(z) = U_s(z) \begin{pmatrix} Z^{-1} & 0 \\ 0 & Z \end{pmatrix} V_r(z) \begin{pmatrix} T_{rs}(Zq^{-1}) & 0 \\ 0 & T_{rs}(Z) \end{pmatrix}.$$

This is equivalent to

$$T_{rs}(q^{-1}Z)(y_s - x_r\mu_r\mu_s Z^2) = T_{rs}(Z)(y_r - x_s\mu_r\mu_s Z^2),$$

which is equivalent to the $T_{rs}$ defining relation (2.9).

(ii) The statement is equivalent to

$$S_{r's}(\chi)\{U_r(z), V_{r'}(z)\} \otimes \{U_s(z), V_{s'}(z)\} = \{U_r(z), V_s(z)\} \otimes \{U_{r'}(z), V_{s'}(z)S_{r's}(\chi)\},$$

where $\chi := X^{-1} \otimes X$ and hence

$$S_{r's}(\chi) \begin{pmatrix} Z_1^{-1} & 0 \\ 0 & Z_1 \end{pmatrix} V_{r'}(z) \begin{pmatrix} \chi^{-1} & 0 \\ 0 & 1 \end{pmatrix} U_s(z) \begin{pmatrix} Z_2^{-1} & 0 \\ 0 & Z_2 \end{pmatrix}$$
$$= \begin{pmatrix} Z_1^{-1} & 0 \\ 0 & Z_1 \end{pmatrix} V_s(z) \begin{pmatrix} \chi^{-1} & 0 \\ 0 & 1 \end{pmatrix} U_{r'}(z) \begin{pmatrix} Z_2^{-1} & 0 \\ 0 & Z_2 \end{pmatrix} S_{r's},$$

which simplifies to

$$\begin{pmatrix} S_{r's}(q^{-1}\chi) & 0 \\ 0 & S_{rs'}(q\chi) \end{pmatrix} V_{r'}(z) \begin{pmatrix} \chi^{-1} & 0 \\ 0 & 1 \end{pmatrix} U_s(z)$$
$$= V_s(z) \begin{pmatrix} \chi^{-1} & 0 \\ 0 & 1 \end{pmatrix} U_{r'}(z) \begin{pmatrix} S_{r's}(q^{-1}\chi) & 0 \\ 0 & S_{rs'}(q\chi) \end{pmatrix}.$$

This is equivalent to the relation

$$S_{r's}(q^{-1}\chi)\mu_s(qx_s - y_{r'}\chi) = S_{r's}(q\chi)\mu_{r'}(qx_{r'} - y_s\chi),$$

which in turn is equivalent to the defining condition (2.10).

(iii) This statement follows directly from (i) and (ii). □

Let us now introduce the standard 6-vertex model R-matrix defined algebraically as the $\widetilde{U}_q(\widehat{\mathfrak{sl}}_2)$ isomorphism

$$\check{R}(z/w) : V_z \otimes V_w \to V_w \otimes V_z.$$

Fixing the arbitrary multiplicative normalization we have[2]

---

[2]This may look like we have swapped $b(z)$ and $c(z)$ compared to usual conventions, but this is simply because we are dealing with $\check{R} = PR$.

$$\check{R}(z) = \begin{pmatrix} a(z) & 0 & 0 & 0 \\ 0 & c(z) & b(z) & 0 \\ 0 & b(z) & c(z) & 0 \\ 0 & 0 & 0 & a(z) \end{pmatrix}, \quad \text{with } (a(z), b(z), c(z)) = \left(1 - q^2 z^2, q(1 - z^2), z(1 - q^2)\right).$$

(3.11)

We use the obvious graphical representation

$$\check{R}(z/w) \ \sim \ z \ \raisebox{-1em}{\rule{0pt}{0pt}} \overset{w}{\vphantom{|}}$$

We then have the following result concerning the commutation of L-operators.

**Proposition 3.11.** *As operators on $W \otimes V \otimes V \to V \otimes V \otimes W$ we have the identities:*

$(i)$  $(\check{R}(z/w) \otimes \mathbb{I})(\mathbb{I} \otimes \check{L}_{\rho_r}(w))(\check{L}_{\rho_r}(z) \otimes \mathbb{I}) = (\mathbb{I} \otimes \check{L}_{\rho_r}(z))(\check{L}_{\rho_r}(w) \otimes \mathbb{I})(\check{R}(z/w) \otimes \mathbb{I}),$

$(ii)$  $(\check{R}(z/w) \otimes \mathbb{I})(\mathbb{I} \otimes \check{L}_{\bar{\rho}_r}(w))(\check{L}_{\bar{\rho}_r}(z) \otimes \mathbb{I}) = (\mathbb{I} \otimes \check{L}_{\bar{\rho}_r}(z))(\check{L}_{\bar{\rho}_r}(w) \otimes \mathbb{I})(\check{R}(z/w) \otimes \mathbb{I}).$

*Proof.* The proof again is either by uniqueness of the associated intertwiners or by explicit checking of the relations. □

The pictures corresponding to Proposition 3.11 are

where graphical composition runs from West to East and North to South as before.

*Remark* 3.12. The approach of [BS90] used L-operators alone: their starting point was to first construct L-operators satisfying

$$(\check{R}(z/w) \otimes \mathbb{I})(\mathbb{I} \otimes \check{L}_{\Omega_{rs}}(w))(\check{L}_{\Omega_{rs}}(z) \otimes \mathbb{I}) = (\mathbb{I} \otimes \check{L}_{\Omega_{rs}}(z))(\check{L}_{\Omega_{rs}}(w) \otimes \mathbb{I})(\check{R}(z/w) \otimes \mathbb{I})$$

and then find an R-matrix satisfying property $(iii)$ of Proposition 3.10. This yielded an R-matrix of the factorized form given by our Equation (2.6).

3.3.1. *Factorization of L-operators.* In the language of L-operators, we have the following analogue of Proposition 3.3.

**Proposition 3.13.** *The L-operators of Equation (3.9) satisfy the following factorization relation:*

$$(\mathbb{I} \otimes \mathcal{O}(\chi)) \, \check{L}_{\Omega_{rs}}(z) \otimes \check{L}_{\varphi} = \check{L}_{\rho_r}(z) \otimes \check{L}_{\bar{\rho}_s}(z) \, (\mathcal{O}(\chi) \otimes \mathbb{I}).$$

Note that the statement in this proposition is represented graphically by

*Proof.* This statement can be rewritten as

$$\mathcal{O}(\chi)\{U_r(z), V_s(z)\} \otimes \{\mathbb{I}, \mathbb{I}\} \quad = \quad \{U_r(z), \mathbb{I}\} \otimes \{V_s(z), \mathbb{I}\}\,\mathcal{O}(\chi)$$

which follows from the more general property

(3.12) $$\mathcal{O}(\chi)\{A, B\} \otimes \{\mathbb{I}, C\} \quad = \quad \{A, \mathbb{I}\} \otimes \{B, C\}\,\mathcal{O}(\chi)$$

where $A, B, C$ are any $N \times N$ complex valued matrices. Property (3.12) is equivalent to

$$\begin{pmatrix} \mathcal{O}(q^{-1}\chi) & 0 \\ 0 & \mathcal{O}(q\chi) \end{pmatrix} B \begin{pmatrix} \chi^{-1} & 0 \\ 0 & 1 \end{pmatrix}$$
$$= \begin{pmatrix} \chi^{-1} & 0 \\ 0 & 1 \end{pmatrix} B \begin{pmatrix} \mathcal{O}(q^{-1}\chi) & 0 \\ 0 & \mathcal{O}(q\chi) \end{pmatrix}.$$

This requirement in turn reduces to $\mathcal{O}(q\chi) = \chi\,\mathcal{O}(q^{-1}\chi)$, which is the defining property (3.1) of $\mathcal{O}(\chi)$. $\qquad\square$

3.3.2. *Fusion of L-operators.* Now we turn to the SES relations of Theorem 3.7, and their manifestation in terms of L-operators.

**Proposition 3.14.** *We have the following equalities*

$(i)\quad (\check{L}_{\rho_s}(w) \otimes \mathbb{I})(\mathbb{I} \otimes \check{R}(z_s/w))(\iota_s \otimes \mathbb{I}) = C_1(s,w)(\mathbb{I} \otimes \iota_s)\check{L}_{\rho_{sq}}(w) : W \otimes V \to V \otimes W \otimes V$

$(ii)\quad (\mathbb{I} \otimes \tau_s)(\check{L}_{\rho_s}(w) \otimes \mathbb{I})(\mathbb{I} \otimes \check{R}(z_s/w)) = C_2(s,w)\,\check{L}_{\rho_{sq^{-1}}}(w)(\tau_s \otimes \mathbb{I}) : W \otimes V \otimes V \to V \otimes W,$

$(iii)\quad (\check{L}_{\bar{\rho}_s}(w) \otimes \mathbb{I})(\mathbb{I} \otimes \check{R}(z_s/w))(\bar{\iota}_s \otimes \mathbb{I}) = C_1(s,w)(\mathbb{I} \otimes \bar{\iota}_s)\check{L}_{\bar{\rho}_{sq}}(w) : W \otimes V \to V \otimes W \otimes V$

$(iv)\quad (\mathbb{I} \otimes \bar{\tau}_s)(\check{L}_{\bar{\rho}_s}(w) \otimes \mathbb{I})(\mathbb{I} \otimes \check{R}(z_s/w)) = C_2(s,w)\,\check{L}_{\bar{\rho}_{sq^{-1}}}(w)(\bar{\tau}_s \otimes \mathbb{I}) : W \otimes V \otimes V \to V \otimes W,$

*where the coefficient are given by*

$$C_1(s,w) = q^{-1}b(z_s/w), \quad C_2(s,w) = qa(z_s/w),$$

*and $z_s^2 = \kappa_0\kappa_1 x_s y_s$ as in Theorem 3.7.*

The proof of Proposition 3.14 is given in Appendix A.2. The meaning of these four relations, which are sometimes referred to as 'bootstrap' relations, becomes more transparent when viewing their corresponding graphical realizations:

We have one final pair of fusion relations, now concerning the composite operator $B_{r';ss'} : \Omega_{rr'} \otimes \Omega_{ss'} \to \Omega_{rs} \otimes \Omega_{s'r'}$ defined in Equation 2.15.

**Proposition 3.15.** *We have the following equalities*

$$(i) \quad (B_{r';ss'} \otimes \mathbb{I})(\mathbb{I} \otimes \check{\mathbf{L}}_{\Omega_{ss'}}(z_{r'}))(\bar{I}_{r'} \otimes \mathbb{I}) = E_1(r', s, s')\,(\mathbb{I} \otimes \bar{I}_{r'})\,B_{r'q;ss'} : W \otimes W \to W \otimes W \otimes V$$

$$(ii) \quad (\mathbb{I} \otimes \bar{T}_{r'})(B_{r',ss'} \otimes \mathbb{I})(\mathbb{I} \otimes \check{\mathbf{L}}_{\Omega_{s,s'}}(z_{r'})) = E_2(r', s, s')\,B_{r'q^{-1},ss'}(\bar{T}_{r'} \otimes \mathbb{I}) : W \otimes V \otimes W \to W \otimes W,$$

*where* $\check{\mathbf{L}}_{\Omega_{ss'}}(z)$ *is the* $\widetilde{U}_q(\widehat{\mathfrak{sl}}_2)$ *intertwiner* $: V_z \otimes \Omega_{ss'} \to \Omega_{ss'} \otimes V_z$ *defined above, and the coefficients are given by*

$$E_1(r', s, s') = \frac{\mu_s(q^2 z_{r'}^2 - z_s^2)(q^2 x_{r'}\mu_{r'}\mu_{s'} - y_{s'})}{\mu_s x_s - \mu_{r'} x_{r'} q^2}, \quad E_2(r', s, s') = \frac{\mu_{s'} q^2 (z_{r'}^2 - z_{s'}^2)(x_{r'}\mu_{r'} - x_s\mu_s)}{y_{s'} - x_{r'}\mu_{r'}\mu_{s'}}.$$

The corresponding pictures for the equalities in Proposition 3.15 are

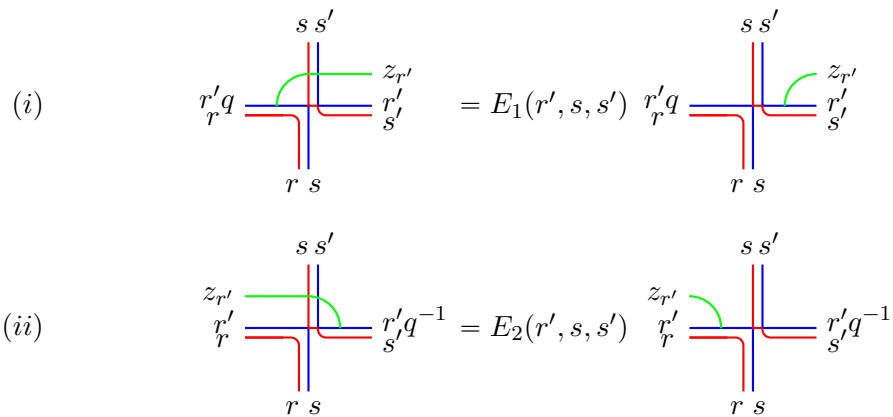

The proof of Proposition 3.15 is given in Appendix A.3.

## 4. Transfer Matrices and Q-operators

In this section we take traces of products of the different L-operators and of $A, B$ defined in Equation (2.15) in order to define and relate transfer matrices and Q-operators acting on quantum spaces which are either $V^{\otimes M}$ or $W^{\otimes M}$. We consider the two cases separately.

### 4.1. Quantum Space $V^{\otimes M}$.

The transfer matrix of the 6-vertex model is defined as follows:

$$(4.1) \qquad T(z) := \mathrm{Tr}_{V^0}\left(R^{0M}(z)\cdots R^{02}(z)R^{01}(z)\right) : V^{\otimes M} \to V^{\otimes M}.$$

Note, that in this definition we use $R = P\check{R}$ for notational simplicity. The superscripts indicate on which pair of spaces in $V^{\otimes M+1} = V^0 \otimes V^1 \otimes \cdots \otimes V^M$ the R-matrix acts. The equivalent graphical realization in terms of $\check{R}$ is simply

$$T(z/w) \sim \quad z \; \text{——|——|——|——|——|——|——}$$

Here, and from now on, it should be understood that there is an implied trace associated with the horizontal auxiliary space in the graphical representation that we use for transfer matrices and Q-operators.

Now we can change the auxiliary representation $V_z$, constructed in terms of the vector space $V$, to any of the four other representations $\rho_r$, $\bar{\rho}$, $\Omega_{rs}$, $\varphi_c$ associated with the vector space $W$. Let us start with $\rho_r$. Then the R-matrix $R(z/w) : V_z \otimes V_w \to V_z \otimes V_w$ appearing appearing in (4.1) is replaced by the corresponding L-operator $L_{\rho_r}(w) : \rho_r \otimes V_w \to \rho_r \otimes V_w$. In this way we define a new operator

$$Q_{\rho_r}(w) := \mathrm{Tr}_W\left(L^M_{\rho_r}(w)\cdots L^2_{\rho_r}(w)L^1_{\rho_r}(w)\right) : V^{\otimes M} \to V^{\otimes M},$$

with $L^i_{\rho_r}$ acting non-trivially on $W \otimes V^i$. The picture is

$$Q_{\rho_r}(w) \sim \quad r$$

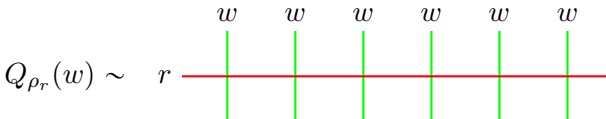

In a similar way we define three more operators $V^{\otimes M} \to V^{\otimes M}$:

$$Q_{\bar{\rho}_r}(w) := \mathrm{Tr}_W \left( L^M_{\bar{\rho}_r}(w) \cdots L^2_{\bar{\rho}_r}(w) L^1_{\bar{\rho}_r}(w) \right) \quad \sim \quad r$$

$$T_{\Omega_{rs}}(w) := \mathrm{Tr}_W \left( L^M_{\Omega_{rs}}(w) \cdots L^2_{\Omega_{rs}}(w) L^1_{\Omega_{rs}}(w) \right) \quad \sim \quad {}^s_r$$

$$T_\varphi := \mathrm{Tr}_W \left( L^M_\varphi \cdots L^2_\varphi L^1_\varphi \right) \quad \sim$$

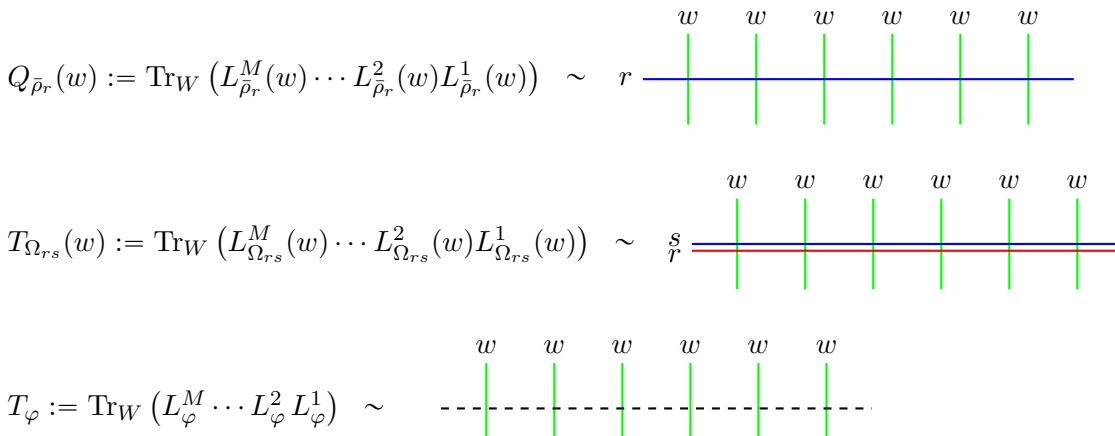

The L-operator $L_\varphi$, given by (3.9), is particularly simple: it is independent of $w$, diagonal and equal to

$$L_\varphi = \begin{pmatrix} q^{-1} Z^{-1} & 0 \\ 0 & Z \end{pmatrix},$$

which immediately gives

$$T_\varphi = q^{-(S_z + M)/2} \mathrm{Tr}_W(Z^{-S_z}), \quad \text{where} \quad S_z = \sum_{i=1}^M \sigma^i_z, \quad \text{with} \quad \sigma_z = \begin{pmatrix} 1 & 0 \\ 0 & -1 \end{pmatrix}.$$

We now present various relations between these T and Q operators that follow directly from the properties of L-operators obtained in Section 3. First of all, we note that the operators $Q_{\rho_r}(w)$, $Q_{\bar{\rho}_r}(w)$ and $T_{\Omega_{rs}}(w)$ do not commute with the total spin operator $S_z$: it is a consequence of the relations $X^N = \mathbb{I}$ that these operators only conserve spin modulo $N$. This feature of cyclic representations was noted in [BS90].

**Proposition 4.1.**

$$[T(z/w), Q_{\rho_r}(w)] = [T(z/w), Q_{\bar{\rho}_r}(w)] = 0.$$

*Proof.* The statement follows from Proposition 3.11 applied to the two auxiliary spaces in each case. $\qquad\square$

**Proposition 4.2.** *We have the factorization property:*

$$T_{\Omega_{rs}(w)} = Q_{\rho_r}(w) Q_{\bar{\rho}_s}(w) T_\varphi^{-1}.$$

*Proof.* This follows directly from Proposition 3.13. The graphical proof is simply

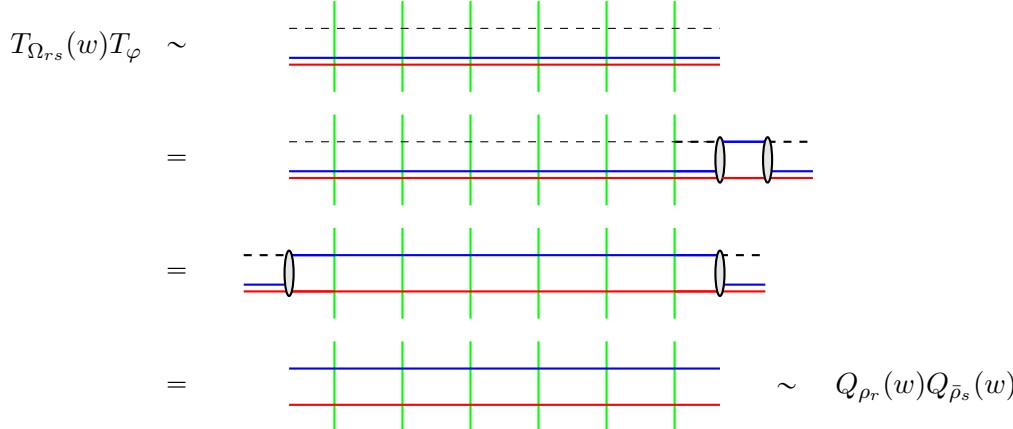

$$T_{\Omega_{rs}}(w)T_\varphi \quad \sim$$

$$\sim \quad Q_{\rho_r}(w)Q_{\bar{\rho}_s}(w)$$

where Proposition 3.13 is used between the second and third diagrams, and cyclicity of the trace is used in the final step. $\qquad\square$

**Proposition 4.3.**

$$(i) \quad Q_{\rho_s}(w)T(z_s/w) = C_1^M(s,w)Q_{\rho_{sq}}(w) + C_2^M(s,w)Q_{\rho_{sq^{-1}}}(w),$$

$$(ii) \quad Q_{\bar{\rho}_s}(w)T(z_s/w) = C_1^M(s,w)Q_{\bar{\rho}_{sq}}(w) + C_2^M(s,w)Q_{\bar{\rho}_{sq^{-1}}}(w).$$

*Proof.* The proof follows from the SESs of Proposition 3.7 $(i)$ and $(ii)$ and Proposition 3.14. We also make use of the standard linear algebra result that given a short exact sequence $0 \to A \to B \to C \to 0$, the trace over $B$ of an operator acting on $B$ can we written as the sum of traces of corresponding operators over $A$ and $C$. Details of this result as applied in this context can be found in [VW20]. $\qquad\square$

Relations $(i)$ and $(ii)$ both contain the same factors $C_1^M(s,w) = (b(z_s/w)/q)^M$ and $C_2^M(s,w) = (qa(z_s/w))^M$. These are precisely the coefficients those that occur in the standard TQ relations of the 6-vertex model - which are in turn used to obtain the Bethe equations (see for example [JMS21]). The apparent difference with the standard TQ relations is the more complicated dependence of the Q operators in (i) and (ii) on $s \in \mathcal{C}_k$ and $w$. However, the TQ relation may be brought into the standard form by making use of the following 'gauge transformation' of the matrices $U_s(w)$ and $V_s(w)$ defined in Equation (3.8):

$$U_s(w) \;=\; w\aleph_s U(z_s/w, \mu_s)\aleph_s^{-1}, \quad \text{where} \quad U(z,\mu) := \begin{pmatrix} 1 & z \\ z & 1 \end{pmatrix}\begin{pmatrix} 1 & 0 \\ 0 & \mu \end{pmatrix} \quad \text{and} \quad \aleph_s := \begin{pmatrix} 1 & 0 \\ 0 & \sqrt{\frac{\kappa_1 y_s}{\kappa_0 x_s}} \end{pmatrix},$$

$$V_s(w) \;=\; w\beth_s V(z_s/w, \mu_s)\beth_s^{-1} \quad \text{where} \quad V(z,\mu) := \begin{pmatrix} 1 & 0 \\ 0 & \mu \end{pmatrix}\begin{pmatrix} -q & z \\ z & -1 \end{pmatrix} \quad \text{and} \quad \beth_s := \begin{pmatrix} 1 & 0 \\ 0 & \sqrt{\frac{\kappa_1 x_s}{\kappa_0 y_s}} \end{pmatrix}.$$

Let us define

$$L(z,\mu) = \{U(z,\mu), \mathbb{I}\}, \quad \bar{L}(z,\mu) = \{V(z,\mu), \mathbb{I}\},$$

using the bracket notation of Equation (3.7). Then it follows that we have

(4.2) $$L_{\rho_s}(w) = w\aleph_s L(z_s/w, \mu_s)\aleph_s^{-1}, \quad L_{\bar{\rho}_s}(w) = w\beth_s \bar{L}(z_s/w, \mu_s)\beth_s^{-1}.$$

Using the transformed L-operators $L(z,\mu)$ and $\bar{L}(z,\mu)$, we can define modified Q operators by

$$Q(z,\mu): \ = \ \underset{W}{\mathrm{Tr}}(L^M(z,\mu)\cdots L^2(z,\mu)L^1(z,\mu)),$$

$$\bar{Q}(z,\mu): \ = \ \underset{W}{\mathrm{Tr}}(\bar{L}^M(z,\mu)\cdots \bar{L}^2(z,\mu)\bar{L}^1(z,\mu)),$$

It is clear, from the form of $U(z,\mu)$ and $V(z,\mu)$, that $Q(z,\mu)$ and $\bar{Q}(z,\mu)$ are polynomial in $z$. It then follows from (4.2) that we have

$$(4.3) \qquad Q(z_s/w,\mu_s) \ = \ w^{-M}\left(\sqrt{\frac{\kappa_1 y_s}{\kappa_0 x_s}}\right)^{\frac{S_z}{2}} Q_{\rho_s}(w)\left(\sqrt{\frac{\kappa_1 y_s}{\kappa_0 x_s}}\right)^{-\frac{S_z}{2}},$$

$$(4.4) \qquad \bar{Q}(z_s/w,\mu_s) \ = \ w^{-M}\left(\sqrt{\frac{\kappa_1 x_s}{\kappa_0 y_s}}\right)^{\frac{S_z}{2}} Q_{\bar{\rho}_s}(w)\left(\sqrt{\frac{\kappa_1 x_s}{\kappa_0 y_s}}\right)^{-\frac{S_z}{2}}.$$

Hence it then follows from Proposition 4.3 and the property $[T(z), S_z] = 0$ that $Q(z,\mu)$ and $\bar{Q}(z,\mu)$ satisfy the standard TQ relations[3]

$$(4.5) \qquad \begin{aligned} Q(z,\mu)T(z) \ &= \ (b(z)/q)^L\, Q(qz,q\mu) + (qa(z))^L\, Q(q^{-1}z,q^{-1}\mu), \\ \bar{Q}(z,\mu)T(z) \ &= \ (b(z)/q)^L\, \bar{Q}(qz,q\mu) + (qa(z))^L\, \bar{Q}(q^{-1}z,q^{-1}\mu). \end{aligned}$$

Before completing this section, we note that $T_{\Omega_{rs}}$ also satisfies the relation

$$(4.6) \qquad T_{\Omega_{rs}}(w)T(z_s/w) = C_1^M(s,w)T_{\Omega_{r,sq}}(w) + C_2^M(s,w)T_{\Omega_{sq^{-1}}}(w).$$

This follows either by the observation that it addition to Proposition 3.14, we also have that

$$(\check{L}_{\Omega rs}(w)\otimes\mathbb{I})(\mathbb{I}\otimes\check{R}(z_s/w))(\bar{I}_s\otimes\mathbb{I}) = C_1(s,w)(\mathbb{I}\otimes\bar{I}_s)\check{L}_{\Omega_{r,sq}}(w) =: W\otimes V \to V\otimes W\otimes V,$$

$$(\mathbb{I}\otimes\bar{T}_s)(\check{L}_{\Omega_{r,s}}(w)\otimes\mathbb{I})(\mathbb{I}\otimes\check{R}(z_s/w)) = C_2(s,w)\,\check{L}_{\Omega_{r,sq^{-1}}}(w)(\bar{T}_s\otimes\mathbb{I}) : W\otimes V\otimes V \to V\otimes W,$$

(which can be proved either by appealing to uniqueness of the corresponding $U_q(\mathfrak{b}_+)$ intertwiner or just by checking the relation), or directly from Propositions 4.2 and 4.3 $(ii)$ plus the observation that $T_\varphi$ commutes with $Q_{\rho_r}$ and $Q_{\bar{\rho}_s}$. Thus we have that $T_{\Omega_{rs}}(w)$ satisfies the same relations (4.6) as the Q operator of the 6-vertex model. This fact was observed in [BS90].

### 4.2. Quantum Space $W^{\otimes M}$.

Now let us define an operator $\mathcal{T}(z): W^{\otimes M} \to W^{\otimes M}$ as follows:

$$\mathcal{T}(z) = \mathrm{Tr}\left(\mathbf{L}^M_{\Omega_{ss'}}(z)\cdots\mathbf{L}^2_{\Omega_{ss'}}(z)\mathbf{L}^1_{\Omega_{ss'}}(z)\right) \quad \sim$$

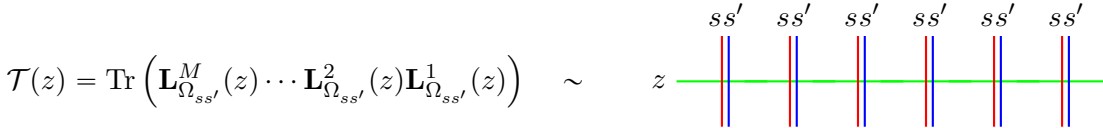

where $\mathbf{L}_{\Omega_{rs}}(z) = P\check{\mathbf{L}}_{\Omega_{rs}}(z)$ is the operator $V\otimes W \to V\otimes W$ defined by 3.10 and given explicitly by Equation (A.2); the superscript $i$ as always indicates that the operator acts on the $i$'th space $W$ in the tensor product $W^{\otimes M}$. The operator $\mathcal{T}(z)$ is the transfer matrix of the $\tau_2$ model as defined in [Bax89, BPAY88, BS90].

---

[3]The dependence of the $Q$ operator on the parameter $\mu$, the algebraic necessity of this parameter, and an explanation how the dependence may be removed is explained in [VW20]

Now we use the operators $B_{r';ss'} = P\check{B}_{r';ss'}$ and $A_{rs;s'} = P\check{A}_{rs;s'}$ specified by Equation 2.15 to define associated operators on $W^{\otimes M+1}$ by

$$
\begin{aligned}
\mathcal{B}_{r';ss'} &= B^{0M}_{r';ss'} \cdots B^{02}_{r';ss'} B^{01}_{r';ss'} \\
\mathcal{A}_{rs;s'} &= A^{0M}_{rs;s'} \cdots A^{02}_{rs;s'} A^{01}_{rs;s'}.
\end{aligned}
$$

Then, we define the following operator $\mathcal{Q}_{r';ss'} : W^{\otimes M} \to W^{\otimes M}$:

$$
\mathcal{Q}_{r';ss'} = \mathrm{Tr}_{W^0}\left(\mathcal{B}_{r';ss'}\right) \sim \quad
$$

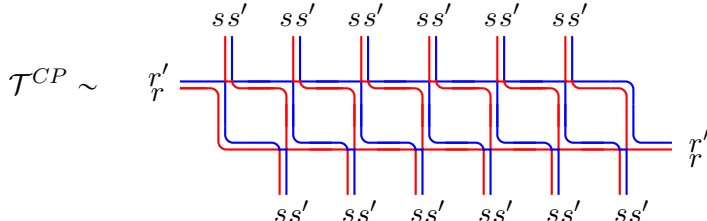

(4.7)

**Proposition 4.4.**

$$
\mathcal{Q}_{r';ss'}\mathcal{T}(z_{r'}) = E_1^M(r';ss')\mathcal{Q}_{r'q;ss'} + E_2^M(r';ss')\mathcal{Q}_{r'q^{-1};ss'}.
$$

*Proof.* The proof follows from the SES of Proposition 3.7 (*iii*) and from Proposition 3.15. $\qquad\square$

In this way, we see that $\mathcal{Q}_{r';ss}$ acts as a Q-operator for the $\tau_2$ model. The connection of this Q-operator with the chiral Potts model is as follows: the transfer matrix of the chiral Potts model is by definition given by

$$
\mathcal{T}^{CP} = \mathop{\mathrm{Tr}}_{W^0}(R(rr';ss')^{0M} \cdots R(rr';ss')^{02} R(rr';ss')^{01}).
$$

It follows from the factorization formula (2.6) that we have

(4.8) $$\mathcal{T}^{CP} = \mathop{\mathrm{Tr}}_{W^0}(\mathcal{A}_{rs;s'} \circ P^{(M)} \circ \mathcal{B}_{r';ss'}),$$

where $P^{(M)} : W^{\otimes M+1} \to W^{\otimes M+1}$ is defined by $P^{(M)}(a \otimes b_1 \otimes b_2 \cdots b_M) = (b_1 \otimes b_2 \cdots b_M \otimes a)$. The corresponding picture is

$$\mathcal{T}^{CP} \sim$$

It was shown in [BS90] that there was a relation between the Q-operator of the $\tau_2$ model (found in [BS90] by explicitly solving of TQ relations) and the half-monodromy matrix of the chiral Potts model. This observation and its algebraic origin are reflected and explained in the current paper by three results: the definition $\mathcal{Q}_{r';ss'} = \mathrm{Tr}_W\left(\mathcal{B}_{r';ss'}\right)$; Proposition 4.4; and the identification (4.8).

## 5. Discussion

In this section we summarize the key results of the paper and comment on their potential use and development. Our first main result is the definition of the two $U_q(\mathfrak{b}_+)$ representations $\rho_r$ and $\bar{\rho}_r$ and the factorization Theorem 3.3. This result is interesting for various reasons: it can be viewed as the analog of the generic-$q$ result (i) mentioned in Section 1; it leads directly to the L-operator factorization of Proposition 3.13 and the transfer matrix factorization of Proposition

4.2; and finally, it allows us, via Proposition 3.6, to characterize chiral Potts Boltzmann weights in an alternative way to Proposition 2.1.

The next key result is the SES/fusion result of Theorem 3.7. This property is manifested in terms of L-operators in the way expressed in Propositions 3.14 and 3.15. These L-operator relations in turn lead to the TQ relations of Propositions 4.3 and 4.4 for operators acting on the quantum spaces $V^{\otimes M}$ and $W^{\otimes M}$. More precisely, when the quantum space is $V^{\otimes M}$ we show that choosing the auxiliary space to be either $\rho$ or $\bar{\rho}$ leads to operators $Q(z, \mu)$ and $\bar{Q}(z, \mu)$ that satisfy the expected TQ relations (4.5) of the 6-vertex model . These operators commute with the 6-vertex model transfer matrix and are polynomial in $z$. Thus they provide a construction of the Q-operators in the $q^N = 1$ case of the 6-vertex model. This is simpler than the construction in the generic-$q$ case that requires an infinite-dimensional auxiliary space as well as regularization of the trace by the introduction of a Cartan element. We also show in Proposition 4.2 that the transfer matrix $T_{\Omega_{rs}}(z)$ factorizes as a product of our two Q-operator, and view this as an analogue of the factorization formula (1.1) of the Verma module transfer matrix in the case of $q$ generic [BLZ97].

For the case when the quantum space is $W^{\otimes M}$, we construct an operator $\mathcal{Q}_{r';ss'}$ by choosing the auxiliary space as $\Omega_{rr'}$. This operator satisfies TQ relations with the transfer matrix $\mathcal{T}(z)$ of the $\tau_2$ model, as expressed in Proposition 4.4. In this case, the shifts in the Q-operator are expressed as $r' \to r'q^{\pm 1}$ which are automorphisms of the curve $\mathcal{C}_k$. Our $\mathcal{Q}_{r';ss'}$ is constructed as a trace over the operator $\mathcal{B}_{r';ss'}$ which is also a building block (in fact one half of the monodromy matrix) of the transfer matrix $\mathcal{T}^{CP}$ of the chiral Potts model as expressed in Equation 4.8.

One chapter of the Q-operator story is missing so far in our discussion of the $q^N$ case: a feature of the generic-$q$ case is that the infinite-dimensional Verma module module $\nu_z^\mu$ mentioned in Section 1 has an infinite-dimensional submodule when $\mu$ takes an integer value $n$. The quotient is then isomorphic to the $n + 1$ dimensional evaluation module (see for example [VW20] for details). This means that the factorization formula (1.1) can be used to express the transfer matrix $T^{(n)}(z)$ associated with the $n + 1$ dimensional auxiliary representation in the form

$$(5.1) \qquad T^{(n)}(z) = \#\mathcal{Q}(zq^{-n/2})\bar{\mathcal{Q}}(zq^{n/2}) - \#\mathcal{Q}(zq^{n/2})\bar{\mathcal{Q}}(zq^{-n/2}).$$

This reducibility does not occur for the general semi-cyclic representations of this paper. In [MLP21] however, L-operators are considered for semi-cyclic representations, and used to define Q-operators that satisfy a relation of the form (5.1). The semi-cyclic limit of our representation $\Omega_{rs}$ was first studied in the paper [IU92], and it would be interesting to attempt to recover the results of [MLP21] from the current paper in this limit.

Both the cyclic representations $\Omega_{rs}$ and the chiral Potts have been generalized to higher-rank cases [DJMM91a, BKMS90]. Also, the generic-$q$ factorization result (i) in Section 1 has been generalized to all untwisted quantum affine algebras in [HJ12, FH15]. It should be possible and would certainly be interesting to generalize our $q^N$ factorization Theorem 3.3 to higher rank.

As mentioned in Section 1, another potential use of our results is in the analysis of open systems. Q-operators for some generic-$q$ open systems have been constructed in terms of infinite-dimensional $U_q(\mathfrak{b}_+)$ representations in [FS15, BT18, VW20, Tsu21, CVW24]. The reason this construction is possible is that the recently formulated universal K-matrix for quantum affine algebras [AV22b, AV22a, AV24] is itself intimately related to the Borel algebras, as well as to coideal subalgebras of the underlying quantum affine algebra. Convergence of the Q-operator is a subtle issue for generic-$q$ open systems; in closed systems, convergence can be ensured by inserting regularizing Cartan elements inside the trace, but this is not possible in open systems as integrability is destroyed by such an insertion. Convergence has to demonstrated for open systems by using convergence properties of $q$-special functions in different regions, and TQ relations must be interpreted in this

light. A key advantage of dealing with $q^N = 1$ is that the $U_q(\mathfrak{b}_+)$ cyclic representations we consider are finite dimensional, and so Q-operators are immediately convergent. Another advantage is that it is simpler to solve reflection equations in the finite rather than infinite-dimensional case for the most general boundary conditions. Thus we expect that the results of the current paper as well as higher-rank generalizations will be a useful tool/test-bed in completing the algebraic description of $Q$ operators for open systems.

**Acknowledgments.** The author thanks Alec Cooper, Bart Vlaar, and Jules Lamers for many useful discussions. He would also like to thank MATRIX and the organizers of the programme 'Mathematics and Physics of Integrability (MPI2024)' during which part of this work was completed. Finally, he acknowledges funding from the EPSRC via grant EP/V008129/1.

## A. Proofs from Section 3

A.1. **Proof of Lemma 3.2.** Using the fact that $q^N = 1$, $\chi^N = \mathbb{I}$, and supposing that $\mathcal{O}^{-1}(\chi) = \sum_{n=0}^{N-1} b_n \chi^N$, we can then write

$$\mathcal{O}(\chi)\mathcal{O}^{-1}(\chi) = (\mathbb{I}, \chi^{N-1}, \chi^{N-2}, \cdots, \chi) A \begin{pmatrix} b_0 \\ b_{N-1} \\ b_{N-2} \\ \vdots \\ b_1 \end{pmatrix}, \quad \text{where } A = \begin{pmatrix} a_0 & a_1 & a_2 & \cdots & a_{N-1} \\ a_{N-1} & a_0 & a_1 & \cdots & a_{N-2} \\ a_{N-2} & a_{N-1} & a_0 & \cdots & a_{N-3} \\ \vdots & \vdots & \vdots & & \vdots \\ a_1 & a_2 & a_3 & \cdots & a_0 \end{pmatrix}.$$

The requirement that $\mathcal{O}(\chi)\mathcal{O}^{-1}(\chi) = \mathbb{I}$ is then equivalent to the matrix relation

(A.1)
$$A \begin{pmatrix} b_0 \\ b_{N-1} \\ b_{N-2} \\ \vdots \\ b_1 \end{pmatrix} = \begin{pmatrix} 1 \\ 0 \\ 0 \\ \vdots \\ 0 \end{pmatrix}.$$

This type of matrix $A$ consisting of cyclically permuted column vectors is know as a *circulant matrix* and has many nice properties (see for example [Dav94]). In particular, its eigenvalues and eigenvectors are known and are respectively

$$\lambda_j = \sum_{n=0}^{N-1} a_n q^{jn} = \mathcal{O}(q^j), \quad v_j = \begin{pmatrix} 1 \\ q^j \\ q^{2j} \\ \vdots \\ q^{(N-1)j} \end{pmatrix}, \quad j \in \{0, 1, \cdots, N-1\}.$$

Hence the matrix $A$ is invertible iff $\mathcal{O}(q^j) \neq 0$ for all $j \in \{0, 1, \cdots, N-1\}$. We find using the Landsberg-Schaar identity that

$$\mathcal{O}(1) = \sum_{n=0}^{N-1} q^{-n^2} = \frac{1 + i^N}{1 + i}\sqrt{N}.$$

The values of $\mathcal{O}(q^j)$ for all $j$ then follow from the required relation $\mathcal{O}(qz) = z\mathcal{O}(q^{-1}z)$ for $z^N = 1$. In this way we see that we indeed have $O(q^j) \neq 0$ for all $j \in \{0, 1, \cdots, N-1\}$ and hence $A$ is invertible. It follows from A.1 that the vector $\underline{b}$ is given by the first column of the matrix $A^{-1}$. $\square$

A.2. **Proof of Proposition 3.14.** There are two approaches to proving (i)-(iv): we can either appeal to uniqueness up to a multiplicative constant of the associated $U_q(\mathfrak{b}_+)$ intertwiner, and thus identify the the two sides by checking a single matrix element; or, we can simply check the full relation as a linear algebra identity. We have done both. As an example, let us show the latter pedestrian approach in detail for the case (iii).

Statement (iii): We can express either side of the identity as a $4 \times 2$ matrix with respect to the space $V$. The left-hand side minus the right-hand side is then

$$
\begin{pmatrix}
\dfrac{\kappa_0 q y_s(-aw+C_1w+cz_s)}{z_s Z} & \dfrac{\kappa_0^2 q y_s^2(b-C_1q)X^{-1}Z}{z_s} \\[2mm]
\dfrac{w(C_1q-b)XZ^{-3}}{q} & -\dfrac{\kappa_0 q y_s\left(-az_s+C_1q^2z+cw\right)}{z_s Z} \\[2mm]
-\dfrac{\mu q XZ^{-1}\left(\kappa_0\kappa_1 x_s y_s\left(C_1q^2-a\right)+cwz_s\right)}{z_s} & \dfrac{\kappa_0\mu_s q w y_s Z(C_1q-b)}{z_s} \\[2mm]
\dfrac{\kappa_1\mu_s x_s(b-C_1q)X^2Z^{-3}}{q} & \dfrac{\mu_s q XZ^{-1}(wz_s(C_1-a)+c\kappa_0\kappa_1 x_s y_s)}{z_s}
\end{pmatrix}
$$

where we have suppressed the arguments of the entries $(a, b, c)$ of the R-matrix $\check{R}(z_s/w)$ given by (3.11), as well as those of $C_1(s, w)$. Recalling that $z_s^2 = \kappa_0\kappa_1 x_s y_s$, we see that each of the terms in the matrix is zero with the choice $C_1(s, w) = q^{-1}b(z_s/w)$.

The proof of Statement (i) is very similar. The proofs of Statements (ii) and (iv) involving showing that all the entries of a $2 \times 4$ are zero with the specified choice of $C_2(s, w) = qa(z_s/w)$.

A.3. **Proof of Proposition 3.15.** Let us define a matrix

$$
\mathcal{M}(z) = \begin{pmatrix}
y_s y_{s'}\kappa_0\kappa_1 Z^{-1} - q^2z^2\mu_s\mu_{s'}Z & qz_s\kappa_0(-y_{s'}Z^{-1} + x_s\mu_s\mu_{s'}Z \\
qz\kappa_1(y_s Z^{-1} - q^2 x_{s'}\mu_s\mu_{s'}Z) & q^2(-z^2 Z^{-1} + x_s x_{s'}\kappa_0\kappa_1\mu_s\mu_{s'}Z)
\end{pmatrix}
$$

Then the explicit expression for the inverse matrix $\check{\mathbf{L}}_{\Omega_{ss'}}(z) : V \otimes W \to W \otimes V$ matrix introduced in Section 3 and involved in Proposition 3.15 is

(A.2) $$
\check{\mathbf{L}}_{\Omega_{ss'}}(z) = \begin{pmatrix}
\mathcal{M}(z)_{00} & X^{-1}\mathcal{M}(z)_{01} \\
X\mathcal{M}(z)_{10} & \mathcal{M}(z)_{11}.
\end{pmatrix}
$$

Each component $\mathcal{M}(z)_{ij}$ then acts $W \to W$.

Statement (i) in the proposition involves two components with respect to the $V$ space. These components are equivalent to the following identities $\mathrm{End}(W \otimes W)$:

(A.3) $\quad d_r B_{r;ss'}(\mathbb{I} \otimes \mathcal{M}(z_r)_{00}) + B_{r;ss'}\chi^{-1}(\mathbb{I} \otimes \mathcal{M}(z_r)_{01}) \quad = \quad d_r E_1(r, s, s')B_{rq;ss'},$

(A.4) $\quad B_{r;ss'}(\mathbb{I} \otimes \mathcal{M}(z_r)_{11}) + d_r B_{r;ss'}\chi^{-1}(\mathbb{I} \otimes \mathcal{M}(z_r)_{10}) \quad = \quad E_1(r, s, s')\chi^{-1}B_{rq;ss'}.$

with

$$
B_{r;ss'} = \sum_{n,m=0}^{N-1} \overline{W}_{rs'}(n)\widehat{W}_{rs}(m)(\mathbb{I} \otimes Z^{2n})\chi^m.
$$

Let us prove (A.3). The simplest way to proceed is to choose a faithful representation of the $X, Z$ algebra on the N-dimensional space $W$. Let us choose $X, Z \in GL_N(\mathbb{C})$ with $X_{ij} = \delta_{i,\mathrm{mod}(j+1,N)}$, $Z_{ij} = \delta_{i,j}q^i$, for $i, j \in \{0, 1, \cdots, N-1\}$. Then each component $\mathcal{M}(z)_{ij}$ is a diagonal matrix with diagonal entries that we denote $\mathcal{M}(z; \ell)_{ij}$ for $\ell \in \{0, 1, \cdots, N-1\}$.

The presumed identity (A.3) then becomes the requirement that

$$\widetilde{W}_{rs'}(\ell + m)\left(d_r\widehat{W}_{rs}(m)\mathcal{M}(z_r;\ell)_{00} + \widehat{W}_{rs}(m+1)\mathcal{M}(z_r;\ell)_{01}\right)$$

(A.5)
$$= \widetilde{W}_{rq,s'}(\ell + m)\widehat{W}_{rq,s}(m)\, d_r E_1(r,s,s')$$

for all $m, \ell \in \{0, \cdots, N-1\}$. Here $\widetilde{W}_{rs}$ is the discrete Fourier transform of $\overline{W}_{rs}$ defined in Equation (2.11. We can re-express the left-hand-side of (A.5) using the recursion relation (2.8 as

(A.6) $\quad \dfrac{\widetilde{W}_{rs'}(\ell + m)\widehat{W}_{rs}(m)}{\mu_s x_s - \mu_r x_r q^{2(m+1)}}\left[d_r(\mu_s x_s - \mu_r x_r q^{2(m+1)})\mathcal{M}(z_r;\ell)_{00} + (\mu_s y_r - \mu_r y_s q^{2m})\mathcal{M}(z_r;\ell)_{01}\right].$

Inserting the explicit expressions for $\mathcal{M}(z;\ell)_{ij}$ the square bracket simplifies to

(A.7) $\qquad [\cdots] = d_r\kappa_0\kappa_1\mu_s q^{-\ell}(q^2 x_r y_r - x_s y_s)(-y_{s'} + q^{2(1+\ell+m)}x_r\mu_r\mu_{s'}).$

Hence our proposition is equivalent to the statement

(A.8) $\quad \dfrac{\widetilde{W}_{rq,s'}(\ell + m)\widehat{W}_{rq,s}(m)}{\widetilde{W}_{r,s'}(\ell + m)\widehat{W}_{r,s}(m)} = \dfrac{\kappa_0\kappa_1\mu_s q^{-\ell}}{E_1(r,s,s')}\dfrac{(q^2 x_r y_r - x_s y_s)(-y_{s'} + q^{2(1+\ell+m)}x_r\mu_r\mu_{s'})}{\mu_s x_s - \mu_r x_r q^{2(m+1)}}$

for all $m, \ell \in \{0, \cdots, N-1\}$.

To prove (A.8) we use induction in $\ell$ and $m$. Firstly, we observe that, with the specified normalizations $\widetilde{W}_{rs}(0) = \widehat{W}_{rs}(0) = 1$, the statement is true for $m = \ell = 0$ when we choose

$$E_1(r,s,s') = \frac{\mu_s(q^2 z_r^2 - z_s^2)(-y_{s'} + q^2 x_r\mu_r\mu_{s'})}{\mu_s x_s - \mu_r x_r q^2}.$$

Now assume that (A.8) is true for a given $(\ell, m)$. Then using (2.13 we find

$$\frac{\widetilde{W}_{rq,s'}(\ell + 1 + m)\widehat{W}_{rq,s}(m)}{\widetilde{W}_{r,s'}(\ell + 1 + m)\widehat{W}_{r,s}(m)} = \frac{\widetilde{W}_{rq,s'}(\ell + m)\widehat{W}_{rq,s}(m)}{\widetilde{W}_{r,s'}(\ell + m)\widehat{W}_{r,s}(m)}\frac{y_{s'} - x_r\mu_r\mu_{s'}q^{2(\ell+2+m)}}{q(y_{s'} - x_r\mu_r\mu_{s'}q^{2(\ell+1+m)})}$$

$$= \frac{\kappa_0\kappa_1\mu_s q^{-(\ell+1)}}{E_1(r,s,s')}\frac{(q^2 x_r y_r - x_s y_s)(-y_{s'} + q^{2(2+\ell+m)}x_r\mu_r\mu_{s'})}{\mu_s x_s - \mu_r x_r q^{2(m+1)}},$$

and thus (A.8) is true for $(\ell + 1, m)$. Similarly, we can use 2.8 to obtain

$$\frac{\widetilde{W}_{rq,s'}(\ell + m + 1)\widehat{W}_{rq,s}(m + 1)}{\widetilde{W}_{r,s'}(\ell + m + 1)\widehat{W}_{r,s}(m + 1)} = \frac{\widetilde{W}_{rq,s'}(\ell + m + 1)\widehat{W}_{rq,s}(m)}{\widetilde{W}_{r,s'}(\ell + m + 1)\widehat{W}_{r,s}(m)}\frac{q(\mu_s x_s - \mu_r x_r q^{2(m+1)})}{\mu_s x_s - \mu_r x_r q^{2(m+2)}}$$

$$= \frac{\kappa_0\kappa_1\mu_s q^{-\ell}}{E_1(r,s,s')}\frac{(q^2 x_r y_r - x_s y_s)(-y_{s'} + q^{2(2+\ell+m)}x_r\mu_r\mu_{s'})}{\mu_s x_s - \mu_r x_r q^{2(m+2)}}.$$

Hence the statement (A.8) is true for $(\ell, m + 1)$. This completes of A.3, and the proof of (A.4) is very similar.

Statement (ii) of Proposition 3.15 is equivalent to the two identities.

(A.9) $\qquad -\chi B_{r,ss'}(\mathbb{I} \otimes \mathcal{M}(z_r)_{00}) + d_r B_{r,ss'}\chi(\mathbb{I} \otimes \mathcal{M}(z_r)_{10}) = -E_2(r,s,s')B_{rq^{-1},ss'}$

(A.10) $\qquad -\chi B_{r,ss'}\chi^{-1}(\mathbb{I} \otimes \mathcal{M}(z_r)_{01}) + d_r B_{r,ss'}(\mathbb{I} \otimes \mathcal{M}(z_r)_{11}) = d_r E_2(r,s,s')B_{rq^{-1},ss'}.$

These can both be proved in a similar way to above. Let us outline the proof of (A.10). The equation is equivalent to the identity

$$\left(-\widetilde{W}_{r,s'}(m+\ell-1)\mathcal{M}(z_r;\ell)_{01} + d_r\widetilde{W}_{r,s'}(m+\ell)\mathcal{M}(z_r;\ell)_{11}\right)\widehat{W}_{rs}(m)$$

(A.11)
$$= d_r E_2(r,s,s')\widetilde{W}_{rq^{-1},s'}(m+\ell)\widehat{W}_{rq^{-1},s}(m).$$

After using the recursion relation (2.13), the left-hand-side of (A.11) simplifies to

$$\left[-(y_r - x_{s'}\mu_r\mu_{s'}q^{2(m+\ell)})\mathcal{M}(z_r;\ell)_{01} + d_r(y_{s'} - x_r\mu_r\mu_{s'}q^{2(m+\ell)})\mathcal{M}(z_r;\ell)_{11}\right]\frac{\widetilde{W}_{r;s'}(m+\ell)\widehat{W}_{rs}(m)}{y_{s'} - x_r\mu_r\mu_{s'}q^{2(m+\ell)}}$$

The square bracket now simplifies to

$$[\cdots] = d_r\mu_{s'}\kappa_0\kappa_1 q^{2+\ell}(x_r y_r - x_{s'}y_{s'})(q^{2m}x_r\mu_r - x_s\mu_s),$$

and Equation (A.10) is equivalent to

$$\frac{\widetilde{W}_{rq^{-1},s'}(m+\ell)\widehat{W}_{rq^{-1},s}(m)}{\widetilde{W}_{r;s'}(m+\ell)\widehat{W}_{rs}(m)} = \frac{\mu_{s'}\kappa_0\kappa_1 q^{2+\ell}(x_r y_r - x_{s'}y_{s'})(q^{2m}x_r\mu_r - x_s\mu_s)}{E_2(r,s,s')(y_{s'} - x_r\mu_r\mu_{s'}q^{2(m+\ell)})}.$$

This statement can be proved by induction in $m$ and $\ell$ as above, with the choice

$$E_2(r,s,s') = \frac{\mu_{s'}q^2(z_r^2 - z_{s'}^2)(x_r\mu_r - x_s\mu_s)}{(y_{s'} - x_r\mu_r\mu_{s'})}.$$

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

*Email address*: r.a.weston@hw.ac.uk