# Peer review of "Cyclic Representations of $U_q(\hat{\mathfrak{sl}}_2)$ and its Borel Subalgebras at Roots of Unity and Q-operators"

_SciPost Physics Core_

## Round 4 · Referee Report · Christian Korff (Referee 1) · 2025-10-6

Strengths

1) A mathematically sound and rigorous construction of Baxter's Q-operator for the XXZ spin-chain at a point which is often neglected and notoriously difficult: the case when the deformation parameter of the quantum group is a root of unity.

2) A development of a graphical calculus which might help readers not familiar with representation theory to understand and appreciate the results.

3) A detailed account of previous results and setting the construction into context with the state of the art on the subject matter.

Weaknesses

The following weaknesses I would consider minor and can be easily addressed by the author:

1) Perhaps due to the technicality of the construction, I think that the physical application and motivation of the construction has been somewhat lost. Given the target audience is in physics, I think it would be good to add a paragraph on this in the introduction and in the conclusion.

Concretely, it is known [Deguchi, Fabricius,McCoy] that the XXZ Hamiltonian exhibits degeneracies in its spectrum at roots of unity and the standard Bethe ansatz construction fails to give an eigenbasis. [See further comments in the report.] It is also known that there are considerable combinatorial aspects/applications (see e.g. [Pronko,Stroganov]) at the root of unity point, which are not mentioned.

2) Large parts of the paper are rather technical (maybe unavoidably so) and require a familiarity with representation theory which the intended readership might not have. In some parts of the preprint the graphical calculus has already been used to make the results more accessible but this could be done more consistently throughout.

Report

The paper introduces finite-dimensional representations of the upper Borel algebra of the affine quantum group for sl(2) at certain [odd?] roots of unity. The latter are then used to give a representation theoretic construction of Baxter's Q-operator and a derivation of functional equations. While previous work on this exists [see Remark 3.8 in the preprint], the representations used in the paper are to the best of my knowledge new and the results go beyond what has previously been known.

Concretely, the representations used in the construction depend on additional parameters which allow one to make contact with generic points on the (higher genus) curve underlying the chiral Potts model. This connection between the XXZ model at roots of unity and the chiral Potts model was first observed by Bazhanov and Stroganov and then further explored algebraically in a series of papers by Date et al. The results in this paper build on this previous work to shed new light on this connection. This application should be emphasised more in my opinion as it is a concrete result of interest to a wider physics community.

My mild criticism is that the conclusion reads as if the representation theoretic construction of the Q-operator has become a goal in its own right and the original physical motivation for it has been somewhat lost: Baxter introduced Q as an "auxiliary matrix" with no direct physical significance. I would wish for a somewhat wider view with some physical motivation included given the journal.

Up to certain minor but essential revisions I am happy to recommed the paper for publication.

Requested changes

It follows a rather long and detailed list of suggested changes. They are made with the best intentions:

[Comments are numbered by Section number and capital letters]

1 Introduction:

1.A. Clarify the comment on the reps $\rho$ and $\bar\rho$ being non-extendable to the full quantum group, as I believe there is now work by Hernandez et al on certain generalisations of affine quantum groups (and shifted versions thereof) where these can be considered as restrictions. This is not a purely technical point, I think in order to understand the "enhanced symmetry" at roots of unity this is important.

1.B. Mention that Eqn 1.2 also holds for $\bar Q$.

1.C. IMPORTANT: Please do explicitly mention in the introduction if your construction applies to even and odd roots of unity. The comment at the start of Section 2 seems to restrict to odd roots of unity only. Previous work has considered both cases.

1.D. In Section 1.2 of the introduction the comment is somewhat misleading giving the reader the impression that no previous construction of Q based on rep theory exists in the literature. Remark 3.8 in the text makes a more accurate statement. Please be as clear in the introduction (including references).

1.E. One of the motivations for previous work on the Q-operator for roots of unity has been the observation by Deguchi et al. that the spectrum of the XXZ Hamiltonian (and the 6-vertex transfer matrix) shows special degeneracies. In particular, the construction in [Kor03a] based on evaluation reps for the full quantum group links these degeneracies to the quantum coadjoint action of de Concini et al. (An infinite-dimensional Lie group action on a 4-dimn'l complex hypersurface; see De Concini C, Kac V and Procesi C 1992 J. Amer. Math. Soc. 5 151-189). The paper [Kor03a] also explains how the points on the hypersurface of de Concini et al correspond to special points on the curve of the chiral Potts model.

The aspect of degeneracies in the spectrum of the XXZ model is completely missing from the discussion and it is currently not clear how your construction helps in understanding the symmetry behind these degeneracies. Could you please comment on this in the introduction and/or the conclusion/discussion. I think it is a rather important aspect for the application of your results and for the target readership of the journal.

1.F. There has been a flurry of developments of combinatorial applications and while it would be beyond the scope of the current paper to cover all of these, it might be possible to at least mention this development starting with the work of Pronko and Stroganov. For example, I would be delighted if your graphical calculus for your reps would open a route to a better understanding of the combinatorics of the chiral Potts model.

1.G. SUGGESTION: Could you please consider inserting a reference table listing the various algebras/reps/symbols/graphic notation either in the introduction or some other prominent/easy to find place in your preprint. This would help the reader enormously to find their way through the sometimes heavy notation. Thank you.

Section 2: Cyclic representations

2.A. From the first paragraph it appears that the entire discussion in your preprint restricts to odd roots of unity. Please make this clear in the introduction, in particular as previous results on Q in the literature include both: even and odd roots of unity.

2.B. The definition of $\tilde U_q(\widehat{sl}_2)$ is not included and some comments should be added about the importance of the additional central elements. I suspect that setting these to 1 gives the "standard" affine quantum group $U_q(\widehat{sl}_2)$ but what does this specialisation mean in terms of the chiral Potts model? A bit more background information for the reader would be helpful here even if you do not include the full definition of either algebra.

2.C. As Section 2 is not self-contained and some definitions/notation have changed from the original reference pease include the precise location when citing , i.e. page/equation number etc. This way the reader can compare more easily.

2.D. Please be consistent when writing intertwiners/modules. In Equation 2.4 the $\check R$-matrix is a map between the module $W$ to itself, below you replace the module $W$ with the symbol $\Omega_{rs}$ which you originally introduce as the rep map. If you want to use notation for a module and its map interchangeably, then please alert the reader to it. [This applies throughout the rest of the preprint where you seem to swap freely between both interpretations.]

2.E. Please state the precise reference for Prop 2.1 (i.e. prop number or page number in the cited reference)

2.F. A reference table listing the symbols/modules/algebra for all the different intertwiners would be most helpful to the reader!

2.G. The graphical notation at the end of 2.1 is motivated by the results in Section 2.2. Maybe comment on this for the benefit of the reader and to motivate your graphical notation.

2.H. Personally, I strongly object to using the same graphical notation for $R$ and $\check R$. The latter should be represented by a 45 deg rotated cross or St. Andrew's cross (like in braid diagrams) which would make drawing pictures easier and more consistent. But the decision is yours.

2.I. The main new objects, the reps $\rho,\bar \rho$, are stated without proof in 2.2. There is neither a proposition environment nor an equation number. I do appreciate that to check that these are indeed reps is tedious but the reader should at least be reassured that this has been done. Please clarify this by stating this either as a Claim or a Proposition.

2.J. I am afraid that I am not willing to believe the claim (at least not on face value) that one can confidently state that these are not restrictions of any root of unity reps for the affine quantum group at roots of unity. Not even sure that these have been classified. Could you please back up your claim or at least tell the reader how you arrived at this conclusion.

Section 3: Factorisation & Fusion

This is by far the most technical part of the paper yet it contains the core material needed to arrive at the functional equations of Section 4 which discusses the physically relevant operators.

3.A. My suggestion is to be ruthless here and follow through with what has in part already been done: state the main results algebraically then pictorially (e.g. Prop 3.11 and then the picture from the proof) and put all the algebraic proofs into the appendix.

I stress that the proofs are essential and should be kept but given the intended readership of the journal it might be best to focus on the main results in the main text and refer the mathematically interested readership to the appendix for details.

Some additional more specific comments:

3.B. After Eqn 3.2 the invertibility of P(z) should be highlighted as Lemma even though I agree that the proof can be omitted.

3.C. Thm 3.7.(ii) is the $T\bar Q$-equation not mentioned in the introduction, please add this as mentioned above. A comment that there are two sets of Bethe equations and solutions would be of interest to the reader. Can anything be said about the polynomial degree of the eigenvalues of $Q$ and $\bar Q$ in a fixed weight space/spin sector to help the reader to appreciate the result?

3.D. Thm 3.7(iii) is closest to the discussion in [Kor03a], the main difference being that here the upper Borel algebra is considered. However, I think Remark 3.8. should maybe acknowledge that this means that the connection to the work of de Concini et al [which should be mentioned as well] is lost and with it the infinite dimn'l group action which acts transitively on the Q-operators and which leaves the transfer matrix fixed. Is there an extension of this action to a larger group?

Section 4: Transfer matrices and Q-operators

This section deals with transfer matrices of different models and this is where maybe the main interest of a physics audience would be. However, it is not made very explicit to the reader and I would strongly suggest that the subsection headings "the XXZ chain"/"the chiral Potts model" would be more appropriate than using the two different modules in the headings.

4.A. I think the importance of the twist parameter mentioned on page 18 in the first paragraph was recognised and made explicit in the works [C.Korff, J.Phys. A37 (2004) 7227-7254 and J. Phys. A: Math. Gen. 38 6641] pre-dating the reference [VW20]. In my opinion this is a misrepresentation of the existing literature.

4.B. I think it is also worthwhile alerting the reader to the fact that a particular choice for the module W has been made when picking the matrices X, Z. There are other description of the module for the full quantum group which allow one to also directly consider the semi-cyclic limit; see [de Concini et al].

4.C. As suggested above, it might be helpful to state each proposition/theorem in this section algebraically and in graphical form. In some instances this just means moving pictures from the proof to the main statement.

4.D. What is missing after Prop 4.3. is a discussion of the resulting Bethe ansatz equations and the implications for the spectrum of the transfer matrix of the XXZ model. This is the physical application and it seems somewhat forgotten.

4.E. Please carefully check the indices in Eqn 4.6, I believe one of the modules in the equation might be missing its second parameter.

4.F. While the connection with the chiral Potts model and the previous results in the literature is made, the discussion is rather terse and ends somewhat abruptly. I would have hoped for at least a remark what new insight the novel construction using the reps of the Borel algebra offers when describing the symmetries and algebraic structure underlying the chiral Potts model. This is not sufficiently commented on in the conclusion/discussion either.

  1. Discussion

The overall criticism is that the focus of the discussion is too much on the representation theoretic construction of the Q-operator and neglects its physical application: the description of the spectrum and its degeneracies of the XXZ transfer matrix and, possibly, the chiral Potts model. In particular, the discussion of the Bethe ansatz and its solutions needs to be mentioned in my opinion.

On the representation theoretic side the work of de Concini et al and the quantum coadjoint action should be reflected on; see my previous comments. There is an infinite dimensional Lie group acting on the set of possible transfer matrices/Q-operators as has been pointed out in the literature [Kor03a]. Are the quantum Borel modules introduced in this article pointing towards a larger symmetry group?

Recommendation

Ask for minor revision

---

## Round 5 · Author Response

List of changes
1 Introduction:
1.A. Clarify the comment on the reps $\rho$ and $\bar{\rho}$ being non-extendable to the full quantum group, as I believe there is now work by Hernandez et al on certain generalisations of affine quantum groups (and shifted versions thereof) where these can be considered as restrictions. This is not a purely technical point, I think in order to understand the "enhanced symmetry" at roots of unity this is important.
R: Point clarified in footnote and references given.
1.B. Mention that Eqn 1.2 also holds for $\widebar{Q}$.
R: Remark 'A similar relation to 1.2 is satisfied by $\widebar{Q}(z)$' added.
1.C. IMPORTANT: Please do explicitly mention in the introduction if your construction applies to even and odd roots of unity. The comment at the start of Section 2 seems to restrict to odd roots of unity only. Previous work has considered both cases.
R: Added: 'Throughout this work we restrict to $N$ odd and $N\geq 3$: the representation theory for $N$ odd and $N$ even has significant differences as explained in ...'
1.D. In Section 1.2 of the introduction the comment is somewhat misleading giving the reader the impression that no previous construction of Q based on rep theory exists in the literature. Remark 3.8 in the text makes a more accurate statement. Please be as clear in the introduction (including references).
R: I have modified the part of Section 1.2 to clarify this point.
1.E. One of the motivations for previous work on the Q-operator for roots of unity has been the observation by Deguchi et al. that the spectrum of the XXZ Hamiltonian (and the 6-vertex transfer matrix) shows special degeneracies. In particular, the construction in [Kor03a] based on evaluation reps for the full quantum group links these degeneracies to the quantum coadjoint action of de Concini et al. (An infinite-dimensional Lie group action on a 4-dimn'l complex hypersurface; see De Concini C, Kac V and Procesi C 1992 J. Amer. Math. Soc. 5 151-189). The paper [Kor03a] also explains how the points on the hypersurface of de Concini et al correspond to special points on the curve of the chiral Potts model.
The aspect of degeneracies in the spectrum of the XXZ model is completely missing from the discussion and it is currently not clear how your construction helps in understanding the symmetry behind these degeneracies. Could you please comment on this in the introduction and/or the conclusion/discussion. I think it is a rather important aspect for the application of your results and for the target readership of the journal.
R: The goals and motivations of the current paper are different and more limited as explained in Section 1.3. I have however, included an extra sentence at the beginning of Section 1.2 to inform readers about the existence and motivation of the earlier work referred to.
1.F. There has been a flurry of developments of combinatorial applications and while it would be beyond the scope of the current paper to cover all of these, it might be possible to at least mention this development starting with the work of Razumov and Stroganov. For example, I would be delighted if your graphical calculus for your reps would open a route to a better understanding of the combinatorics of the chiral Potts model.
R: While I appreciate the suggestion to advertise this work to the combinatorics and integrability community, I think it would be a little misleading to claim a direct connection; to my knowledge the combinatorial connections come from the combinatorial nature of the R-matrices of generic-q representations at roots of unity rather than the representations considered in this paper.
1.G. SUGGESTION: Could you please consider inserting a reference table listing the various algebras/reps/symbols/graphic notation either in the introduction or some other prominent/easy to find place in your preprint. This would help the reader enormously to find their way through the sometimes heavy notation. Thank you.
R: Thanks for the very useful suggestion. I have now included this and I hope it will help to make the paper more accessible.
Section 2: Cyclic representations
2.A. From the first paragraph it appears that the entire discussion in your preprint restricts to odd roots of unity. Please make this clear in the introduction, in particular as previous results on Q in the literature include both: even and odd roots of unity.
R: Done in Sections 1 and 2 and the table.
2.B. The definition of $\tilde{U}_q(\widehat{\mathfrak{sl}}_2)$
is not included and some comments should be added about the importance of the additional central elements. I suspect that setting these to 1 gives the "standard" affine quantum group $U_q(\widehat{\mathfrak{sl}}_2)$
but what does this specialisation mean in terms of the chiral Potts model? A bit more background information for the reader would be helpful here even if you do not include the full definition of either algebra.
R: I have added footnote 3) to explain the origin of the additional central elements at the point I discuss chiral Potts weights. The comultiplication depends on $z_0,z_1$, and setting $z_0,z_1=1$ would move us away from the standard CP weights.
2.C. As Section 2 is not self-contained and some definitions/notation have changed from the original reference pease include the precise location when citing , i.e. page/equation number etc. This way the reader can compare more easily.
R: Done
2.D. Please be consistent when writing intertwiners/modules. In Equation 2.4 the R-matrix is a map between the module $W$ to itself, below you replace the module $W$ with the symbol $\Omega_{rs}$
which you originally introduce as the rep map. If you want to use notation for a module and its map interchangeably, then please alert the reader to it. [This applies throughout the rest of the preprint where you seem to swap freely between both interpretations.]
R: I have now alerted the reader to the useage and meaning of my `abbreviated' notation. See the paragraph beginning 'As we shall be discussing ...' I do want to keep this notation - it is particularly useful for diagrammatics. Hopefully, with the explanatory information now included this will not be confusing.
2.E. Please state the precise reference for Prop 2.1 (i.e. prop number or page number in the cited reference)
R: Done
2.F. A reference table listing the symbols/modules/algebra for all the different intertwiners would be most helpful to the reader!
R: I feel your pain - done.
2.G. The graphical notation at the end of 2.1 is motivated by the results in Section 2.2. Maybe comment on this for the benefit of the reader and to motivate your graphical notation.
R: I have the sentence `To help picture the various intertwiners, and for later usage it, we introduce a graphical notation ...'
2.H. Personally, I strongly object to using the same graphical notation for $R$ and $\check{R}$.
The latter should be represented by a 45 deg rotated cross or St. Andrew's cross (like in braid diagrams) which would make drawing pictures easier and more consistent. But the decision is yours.
R: I thank you for your feedback, but I will retain my St George's cross.
2.I. The main new objects, the reps $\rho$ and $\bar{\rho}$ , are stated without proof in 2.2. There is neither a proposition environment nor an equation number. I do appreciate that to check that these are indeed reps is tedious but the reader should at least be reassured that this has been done. Please clarify this by stating this either as a Claim or a Proposition.
R: Proposition 2.2 now added.
2.J. I am afraid that I am not willing to believe the claim (at least not on face value) that one can confidently state that these are not restrictions of any root of unity reps for the affine quantum group at roots of unity. Not even sure that these have been classified. Could you please back up your claim or at least tell the reader how you arrived at this conclusion.
R: Stated and proved in Prop 2.3.
Section 3: Factorisation & Fusion
This is by far the most technical part of the paper yet it contains the core material needed to arrive at the functional equations of Section 4 which discusses the physically relevant operators.
3.A. My suggestion is to be ruthless here and follow through with what has in part already been done: state the main results algebraically then pictorially (e.g. Prop 3.11 and then the picture from the proof) and put all the algebraic proofs into the appendix.
I stress that the proofs are essential and should be kept but given the intended readership of the journal it might be best to focus on the main results in the main text and refer the mathematically interested readership to the appendix for details.
R: After further reflection I think that the level to which I have used diagrammatics is already a good compromise: I think there is enough to give the reader an understanding of key results such as Proposition 3.15 and 3.16, but not too much to snow them under, which I think might be the case if I systematically introduced diagrammatics for every result such as Proposition 3.11. So, I appreciate the suggestion but I would prefer to leave things as they are.
Some additional more specific comments:
3.B. After Eqn 3.2 the invertibility of P(z) should be highlighted as Lemma even though I agree that the proof can be omitted.
R: New Lemma 3.6 included.
3.C. Thm 3.7.(ii) is the $T \bar{Q}$ equation not mentioned in the introduction, please add this as mentioned above. A comment that there are two sets of Bethe equations and solutions would be of interest to the reader. Can anything be said about the polynomial degree of the eigenvalues of $Q$ and $\bar{Q}$ in a fixed weight space/spin sector to help the reader to appreciate the result?
R: The $T \bar{Q}$ equation has now been mentioned in the Introduction. It also appears explicitly in Prop. 4.3 (ii) and Equation (4.5).
3.D. Thm 3.7(iii) is closest to the discussion in [Kor03a], the main difference being that here the upper Borel algebra is considered. However, I think Remark 3.8. should maybe acknowledge that this means that the connection to the work of de Concini et al [which should be mentioned as well] is lost and with it the infinite dimn'l group action which acts transitively on the Q-operators and which leaves the transfer matrix fixed. Is there an extension of this action to a larger group?
R: I have mentioned the quantum coadjoint action at the end of Section 4.1. I have alsonow referred to and cited De Concini et al in Sections 1 and 2.
Section 4: Transfer matrices and Q-operators
This section deals with transfer matrices of different models and this is where maybe the main interest of a physics audience would be. However, it is not made very explicit to the reader and I would strongly suggest that the subsection headings "the XXZ chain"/"the chiral Potts model" would be more appropriate than using the two different modules in the headings.
R: Subsection headings changed.
4.A. I think the importance of the twist parameter mentioned on page 18 in the first paragraph was recognised and made explicit in the works [C.Korff, J.Phys. A37 (2004) 7227-7254 and J. Phys. A: Math. Gen. 38 6641] pre-dating the reference [VW20]. In my opinion this is a misrepresentation of the existing literature.
R: No misrepresentation was intended. Now changed.
4.B. I think it is also worthwhile alerting the reader to the fact that a particular choice for the module W has been made when picking the matrices X, Z. There are other description of the module for the full quantum group which allow one to also directly consider the semi-cyclic limit; see [de Concini et al].
R: I have emphasised that the $X,Z$ representation given is one particular choice.
4.C. As suggested above, it might be helpful to state each proposition/theorem in this section algebraically and in graphical form. In some instances this just means moving pictures from the proof to the main statement.
R: See earlier comment on the use of diagrammatics.
4.D. What is missing after Prop 4.3. is a discussion of the resulting Bethe ansatz equations and the implications for the spectrum of the transfer matrix of the XXZ model. This is the physical application and it seems somewhat forgotten.
R: As mentioned above, the goals and motivations of the current paper are more limited - to fill a gap in the existing algebraic picture. I have nothing useful to say about implications for the transfer matrix spectrum at this point.
4.E. Please carefully check the indices in Eqn 4.6, I believe one of the modules in the equation might be missing its second parameter.
R: Thanks - additional parameter added.
4.F. While the connection with the chiral Potts model and the previous results in the literature is made, the discussion is rather terse and ends somewhat abruptly. I would have hoped for at least a remark what new insight the novel construction using the reps of the Borel algebra offers when describing the symmetries and algebraic structure underlying the chiral Potts model. This is not sufficiently commented on in the conclusion/discussion either.
Discussion
The overall criticism is that the focus of the discussion is too much on the representation theoretic construction of the Q-operator and neglects its physical application: the description of the spectrum and its degeneracies of the XXZ transfer matrix and, possibly, the chiral Potts model. In particular, the discussion of the Bethe ansatz and its solutions needs to be mentioned in my opinion.
On the representation theoretic side the work of de Concini et al and the quantum coadjoint action should be reflected on; see my previous comments. There is an infinite dimensional Lie group acting on the set of possible transfer matrices/Q-operators as has been pointed out in the literature [Kor03a]. Are the quantum Borel modules introduced in this article pointing towards a larger symmetry group?
R: This is indeed a paper that concentrates on the representation theory - most importantly the two Theorems 3.3 and 3.6. These theorems lead to new expressions for Q-operators satisfying the TQ relations and the new factorization property of Proposition 4.2. My personal main interest is to use these developments in the analysis of open systems. I have now cited the earlier work on the symmetries of the transfer spectrum and the quantum coadjoint action. I hope that those with more expertise in the Transfer matrix/Bethe Ansatz spectrum will be able to exploit my algebraic construction to investigate these applications further.

Anonymous on 2025-12-15 [id 6148]
My apologies that the syntax of one response is garbled. It should be as follows:
2.D. Please be consistent when writing intertwiners/modules. In Equation 2.4 the R-matrix is a map between the module $W$ to itself, below you replace the module$W$= with the symbol $\Omega_{rs}$ which you originally introduce as the rep map. If you want to use notation for a module and its map interchangeably, then please alert the reader to it. [This applies throughout the rest of the preprint where you seem to swap freely between both interpretations.]
R: I have now alerted the reader to the useage and meaning of my `abbreviated' notation. See the paragraph beginning 'As we shall be discussing ...' I do want to keep this notation - it is particularly useful for diagrammatics. Hopefully, with the explanatory information now included this will not be confusing.

---

## Round 5 · List of Changes

1 Introduction:
1.A. Clarify the comment on the reps $\rho$ and $\bar{\rho}$ being non-extendable to the full quantum group, as I believe there is now work by Hernandez et al on certain generalisations of affine quantum groups (and shifted versions thereof) where these can be considered as restrictions. This is not a purely technical point, I think in order to understand the "enhanced symmetry" at roots of unity this is important.
R: Point clarified in footnote and references given.
1.B. Mention that Eqn 1.2 also holds for $\widebar{Q}$.
R: Remark 'A similar relation to 1.2 is satisfied by $\widebar{Q}(z)$' added.
1.C. IMPORTANT: Please do explicitly mention in the introduction if your construction applies to even and odd roots of unity. The comment at the start of Section 2 seems to restrict to odd roots of unity only. Previous work has considered both cases.
R: Added: 'Throughout this work we restrict to $N$ odd and $N\geq 3$: the representation theory for $N$ odd and $N$ even has significant differences as explained in ...'
1.D. In Section 1.2 of the introduction the comment is somewhat misleading giving the reader the impression that no previous construction of Q based on rep theory exists in the literature. Remark 3.8 in the text makes a more accurate statement. Please be as clear in the introduction (including references).
R: I have modified the part of Section 1.2 to clarify this point.
1.E. One of the motivations for previous work on the Q-operator for roots of unity has been the observation by Deguchi et al. that the spectrum of the XXZ Hamiltonian (and the 6-vertex transfer matrix) shows special degeneracies. In particular, the construction in [Kor03a] based on evaluation reps for the full quantum group links these degeneracies to the quantum coadjoint action of de Concini et al. (An infinite-dimensional Lie group action on a 4-dimn'l complex hypersurface; see De Concini C, Kac V and Procesi C 1992 J. Amer. Math. Soc. 5 151-189). The paper [Kor03a] also explains how the points on the hypersurface of de Concini et al correspond to special points on the curve of the chiral Potts model.
The aspect of degeneracies in the spectrum of the XXZ model is completely missing from the discussion and it is currently not clear how your construction helps in understanding the symmetry behind these degeneracies. Could you please comment on this in the introduction and/or the conclusion/discussion. I think it is a rather important aspect for the application of your results and for the target readership of the journal.
R: The goals and motivations of the current paper are different and more limited as explained in Section 1.3. I have however, included an extra sentence at the beginning of Section 1.2 to inform readers about the existence and motivation of the earlier work referred to.
1.F. There has been a flurry of developments of combinatorial applications and while it would be beyond the scope of the current paper to cover all of these, it might be possible to at least mention this development starting with the work of Razumov and Stroganov. For example, I would be delighted if your graphical calculus for your reps would open a route to a better understanding of the combinatorics of the chiral Potts model.
R: While I appreciate the suggestion to advertise this work to the combinatorics and integrability community, I think it would be a little misleading to claim a direct connection; to my knowledge the combinatorial connections come from the combinatorial nature of the R-matrices of generic-q representations at roots of unity rather than the representations considered in this paper.
1.G. SUGGESTION: Could you please consider inserting a reference table listing the various algebras/reps/symbols/graphic notation either in the introduction or some other prominent/easy to find place in your preprint. This would help the reader enormously to find their way through the sometimes heavy notation. Thank you.
R: Thanks for the very useful suggestion. I have now included this and I hope it will help to make the paper more accessible.
Section 2: Cyclic representations
2.A. From the first paragraph it appears that the entire discussion in your preprint restricts to odd roots of unity. Please make this clear in the introduction, in particular as previous results on Q in the literature include both: even and odd roots of unity.
R: Done in Sections 1 and 2 and the table.
2.B. The definition of $\tilde{U}_q(\widehat{\mathfrak{sl}}_2)$
is not included and some comments should be added about the importance of the additional central elements. I suspect that setting these to 1 gives the "standard" affine quantum group $U_q(\widehat{\mathfrak{sl}}_2)$
but what does this specialisation mean in terms of the chiral Potts model? A bit more background information for the reader would be helpful here even if you do not include the full definition of either algebra.
R: I have added footnote 3) to explain the origin of the additional central elements at the point I discuss chiral Potts weights. The comultiplication depends on $z_0,z_1$, and setting $z_0,z_1=1$ would move us away from the standard CP weights.
2.C. As Section 2 is not self-contained and some definitions/notation have changed from the original reference pease include the precise location when citing , i.e. page/equation number etc. This way the reader can compare more easily.
R: Done
2.D. Please be consistent when writing intertwiners/modules. In Equation 2.4 the R-matrix is a map between the module $W$ to itself, below you replace the module $W$ with the symbol $\Omega_{rs}$
which you originally introduce as the rep map. If you want to use notation for a module and its map interchangeably, then please alert the reader to it. [This applies throughout the rest of the preprint where you seem to swap freely between both interpretations.]
R: I have now alerted the reader to the useage and meaning of my `abbreviated' notation. See the paragraph beginning 'As we shall be discussing ...' I do want to keep this notation - it is particularly useful for diagrammatics. Hopefully, with the explanatory information now included this will not be confusing.
2.E. Please state the precise reference for Prop 2.1 (i.e. prop number or page number in the cited reference)
R: Done
2.F. A reference table listing the symbols/modules/algebra for all the different intertwiners would be most helpful to the reader!
R: I feel your pain - done.
2.G. The graphical notation at the end of 2.1 is motivated by the results in Section 2.2. Maybe comment on this for the benefit of the reader and to motivate your graphical notation.
R: I have the sentence `To help picture the various intertwiners, and for later usage it, we introduce a graphical notation ...'
2.H. Personally, I strongly object to using the same graphical notation for $R$ and $\check{R}$.
The latter should be represented by a 45 deg rotated cross or St. Andrew's cross (like in braid diagrams) which would make drawing pictures easier and more consistent. But the decision is yours.
R: I thank you for your feedback, but I will retain my St George's cross.
2.I. The main new objects, the reps $\rho$ and $\bar{\rho}$ , are stated without proof in 2.2. There is neither a proposition environment nor an equation number. I do appreciate that to check that these are indeed reps is tedious but the reader should at least be reassured that this has been done. Please clarify this by stating this either as a Claim or a Proposition.
R: Proposition 2.2 now added.
2.J. I am afraid that I am not willing to believe the claim (at least not on face value) that one can confidently state that these are not restrictions of any root of unity reps for the affine quantum group at roots of unity. Not even sure that these have been classified. Could you please back up your claim or at least tell the reader how you arrived at this conclusion.
R: Stated and proved in Prop 2.3.
Section 3: Factorisation & Fusion
This is by far the most technical part of the paper yet it contains the core material needed to arrive at the functional equations of Section 4 which discusses the physically relevant operators.
3.A. My suggestion is to be ruthless here and follow through with what has in part already been done: state the main results algebraically then pictorially (e.g. Prop 3.11 and then the picture from the proof) and put all the algebraic proofs into the appendix.
I stress that the proofs are essential and should be kept but given the intended readership of the journal it might be best to focus on the main results in the main text and refer the mathematically interested readership to the appendix for details.
R: After further reflection I think that the level to which I have used diagrammatics is already a good compromise: I think there is enough to give the reader an understanding of key results such as Proposition 3.15 and 3.16, but not too much to snow them under, which I think might be the case if I systematically introduced diagrammatics for every result such as Proposition 3.11. So, I appreciate the suggestion but I would prefer to leave things as they are.
Some additional more specific comments:
3.B. After Eqn 3.2 the invertibility of P(z) should be highlighted as Lemma even though I agree that the proof can be omitted.
R: New Lemma 3.6 included.
3.C. Thm 3.7.(ii) is the $T \bar{Q}$ equation not mentioned in the introduction, please add this as mentioned above. A comment that there are two sets of Bethe equations and solutions would be of interest to the reader. Can anything be said about the polynomial degree of the eigenvalues of $Q$ and $\bar{Q}$ in a fixed weight space/spin sector to help the reader to appreciate the result?
R: The $T \bar{Q}$ equation has now been mentioned in the Introduction. It also appears explicitly in Prop. 4.3 (ii) and Equation (4.5).
3.D. Thm 3.7(iii) is closest to the discussion in [Kor03a], the main difference being that here the upper Borel algebra is considered. However, I think Remark 3.8. should maybe acknowledge that this means that the connection to the work of de Concini et al [which should be mentioned as well] is lost and with it the infinite dimn'l group action which acts transitively on the Q-operators and which leaves the transfer matrix fixed. Is there an extension of this action to a larger group?
R: I have mentioned the quantum coadjoint action at the end of Section 4.1. I have alsonow referred to and cited De Concini et al in Sections 1 and 2.
Section 4: Transfer matrices and Q-operators
This section deals with transfer matrices of different models and this is where maybe the main interest of a physics audience would be. However, it is not made very explicit to the reader and I would strongly suggest that the subsection headings "the XXZ chain"/"the chiral Potts model" would be more appropriate than using the two different modules in the headings.
R: Subsection headings changed.
4.A. I think the importance of the twist parameter mentioned on page 18 in the first paragraph was recognised and made explicit in the works [C.Korff, J.Phys. A37 (2004) 7227-7254 and J. Phys. A: Math. Gen. 38 6641] pre-dating the reference [VW20]. In my opinion this is a misrepresentation of the existing literature.
R: No misrepresentation was intended. Now changed.
4.B. I think it is also worthwhile alerting the reader to the fact that a particular choice for the module W has been made when picking the matrices X, Z. There are other description of the module for the full quantum group which allow one to also directly consider the semi-cyclic limit; see [de Concini et al].
R: I have emphasised that the $X,Z$ representation given is one particular choice.
4.C. As suggested above, it might be helpful to state each proposition/theorem in this section algebraically and in graphical form. In some instances this just means moving pictures from the proof to the main statement.
R: See earlier comment on the use of diagrammatics.
4.D. What is missing after Prop 4.3. is a discussion of the resulting Bethe ansatz equations and the implications for the spectrum of the transfer matrix of the XXZ model. This is the physical application and it seems somewhat forgotten.
R: As mentioned above, the goals and motivations of the current paper are more limited - to fill a gap in the existing algebraic picture. I have nothing useful to say about implications for the transfer matrix spectrum at this point.
4.E. Please carefully check the indices in Eqn 4.6, I believe one of the modules in the equation might be missing its second parameter.
R: Thanks - additional parameter added.
4.F. While the connection with the chiral Potts model and the previous results in the literature is made, the discussion is rather terse and ends somewhat abruptly. I would have hoped for at least a remark what new insight the novel construction using the reps of the Borel algebra offers when describing the symmetries and algebraic structure underlying the chiral Potts model. This is not sufficiently commented on in the conclusion/discussion either.
Discussion
The overall criticism is that the focus of the discussion is too much on the representation theoretic construction of the Q-operator and neglects its physical application: the description of the spectrum and its degeneracies of the XXZ transfer matrix and, possibly, the chiral Potts model. In particular, the discussion of the Bethe ansatz and its solutions needs to be mentioned in my opinion.
On the representation theoretic side the work of de Concini et al and the quantum coadjoint action should be reflected on; see my previous comments. There is an infinite dimensional Lie group acting on the set of possible transfer matrices/Q-operators as has been pointed out in the literature [Kor03a]. Are the quantum Borel modules introduced in this article pointing towards a larger symmetry group?
R: This is indeed a paper that concentrates on the representation theory - most importantly the two Theorems 3.3 and 3.6. These theorems lead to new expressions for Q-operators satisfying the TQ relations and the new factorization property of Proposition 4.2. My personal main interest is to use these developments in the analysis of open systems. I have now cited the earlier work on the symmetries of the transfer spectrum and the quantum coadjoint action. I hope that those with more expertise in the Transfer matrix/Bethe Ansatz spectrum will be able to exploit my algebraic construction to investigate these applications further.

Anonymous on 2025-12-15 [id 6148]
My apologies that the syntax of one response is garbled. It should be as follows:
2.D. Please be consistent when writing intertwiners/modules. In Equation 2.4 the R-matrix is a map between the module $W$ to itself, below you replace the module$W$= with the symbol $\Omega_{rs}$ which you originally introduce as the rep map. If you want to use notation for a module and its map interchangeably, then please alert the reader to it. [This applies throughout the rest of the preprint where you seem to swap freely between both interpretations.]
R: I have now alerted the reader to the useage and meaning of my `abbreviated' notation. See the paragraph beginning 'As we shall be discussing ...' I do want to keep this notation - it is particularly useful for diagrammatics. Hopefully, with the explanatory information now included this will not be confusing.

---

## Editorial Decision

unknown